# The role of calcium in regulating marine phosphorus burial and atmospheric oxygenation

Mingyu Zhao[1✉], Shuang Zhang [1], Lidya G. Tarhan[1], Christopher T. Reinhard[2] & Noah Planavsky[1✉]

The marine phosphorus cycle plays a critical role in controlling the extent of global primary productivity and thus atmospheric $p\mathrm{O}_2$ on geologic time scales. However, previous attempts to model carbon–phosphorus-oxygen feedbacks have neglected key parameters that could shape the global P cycle. Here we present new diagenetic models to fully parameterize marine P burial. We have also coupled this diagenetic framework to a global carbon cycle model. We find that seawater calcium concentration, by strongly influencing carbonate fluorapatite (CFA) formation, is a key factor controlling global phosphorus cycling, and therefore plays a critical role in shaping the global oxygen cycle. A compilation of Cenozoic deep-sea sedimentary phosphorus speciation data provides empirical support for the idea that CFA formation is strongly influenced by marine Ca concentrations. Therefore, we propose a previously overlooked coupling between Phanerozoic tectonic cycles, the major-element composition of seawater, the marine phosphorus cycle, and atmospheric $p\mathrm{O}_2$.

---

[1] Department of Geology and Geophysics, Yale University, 210 Whitney Ave, New Haven, CT 06511, USA. [2] School of Earth and Atmospheric Sciences, Georgia Institute of Technology, GA 30332, USA. ✉email: mingyu.zhao@yale.edu; noah.planavsky@yale.edu

Although there are few constraints on the how the size of the biosphere has changed through Earth's history, phosphorus (P) is commonly considered to be the major limiting nutrient for primary production on geologic time scales. It is also generally accepted that P has a critical role in regulating the extent of organic carbon (C) burial and thus atmospheric oxygen levels ($pO_2$)[1–7]. However, our understanding of the factors that govern the retention or regeneration of P during burial and subsequent chemical alteration of marine sediments (diagenesis) arguably remains limited. Traditionally, bottom-water oxygen concentrations have been assumed to exert the largest influence on P burial[4]. More recently, it has been suggested that the rise of bioturbation in the early Paleozoic significantly increased marine P burial[6,8]. However, although these factors may influence the burial of organic and iron-bound P phases, the dominant controls on the formation of carbonate fluorapatite (CFA)—the largest modern authigenic P burial flux[9]—are relatively poorly understood. As CFA accounts for >50% of marine P burial[9], this translates into considerable uncertainty regarding the chief factors responsible for global P cycling and, by extension, the processes regulating global organic carbon burial and atmospheric $pO_2$ on geologic time scales.

The factors regulating CFA formation and burial in marine sediments and the relative importance of environmental and biological boundary conditions in influencing this process have been underexplored. For example, the kinetics of CFA formation are typically simplified for use in diagenetic models; commonly, CFA formation is parameterized as a linear relationship with dissolved phosphate (e.g., refs. [5,8,10]), and the roles played by other key CFA components such as $Ca^{2+}$, $CO_3^{2-}$, and $F^-$ are largely overlooked. In particular, calcium (Ca) is the most abundant ionic species in CFA ($Ca_{9.54}Na_{0.33}Mg_{0.13}(PO_4)_{4.8}(CO_3)_{1.2}F_{2.48}$, ref. [11]) and seawater dissolved Ca is the single most important Ca source for CFA formation in the upper marine sediment pile. Intuitively, the large swings in seawater Ca concentration during the Phanerozoic[12–14] could have significantly influenced the saturation state and thus precipitation rate of CFA. Here, using models and empirical data, we make a case that seawater Ca concentrations are a key factor shaping marine P burial. Our model results also suggest a significant role for seawater Ca on the evolution of Earth's redox budget.

## Results and discussion

**Model evidence for a Ca control on marine P burial.** To explore the idea that Ca may exercise a major control on P cycling, we first built a simple diagenetic model that includes only organic matter, organic P, $Ca^{2+}$, $CO_3^{2-}$, $PO_4^{3-}$, $F^-$, and CFA (see Methods). In contrast to the simplified rate law used in previous studies[5,8,10], this model includes a rate law that describes the formation rate of CFA as a function of its saturation state, which is consistent with experimental observations[15,16]. This exercise indicates that an increase in seawater Ca concentrations will drive a significant increase in CFA formation (Fig. 1 and Supplementary Figure 1), decreasing phosphate diffusion (i.e., P recycling) from the sediment pile back to seawater. In other words, this exercise confirms that Ca promotes CFA formation, which leads to greater P retention in marine sediments (Fig. 1 and Supplementary Figure 1).

To further explore the role played by seawater Ca concentration and determine the relative influence of a range of environmental and biotic factors on P burial in marine systems, we also developed an extended, multicomponent reaction-transport marine sediment diagenesis model, which includes over 30 diagenetic reactions, a saturation state-dependent rate law for CFA formation, and complete, detailed parameterization of

porewater pH, carbonate chemistry, and adsorption (see Methods and Supplementary Tables 1–5). We have also included the iron phosphate mineral vivianite (which is a significant P sink in some restricted marine settings today[8]). We calibrated our CFA rate law in our fully parameterized model using sedimentary porewater and solid-phase data collected from a well-characterized modern shallow marine site (ref. [3], Friends of Anoxic Mud site, FOAM) and the deep-sea ODP site 1226 (Supplementary Figures 2–3, see Supplementary Methods for further description). Using this extended model, we then explored the relative influence of a range of environmental and biotic factors on P burial.

Our model results show that the intensity of bioturbation, the magnitude of the organic matter flux to the sediment surface, and the concentration of dissolved oxygen in bottom waters all have non-negligible and nonlinear effects on CFA formation and total P burial (Fig. 2 and Supplementary Figures 4–6). Our model, like other recently developed models[8], also indicates that increases in bioturbation intensity and depth can be associated with increases in CFA and total P burial, although enhancement of P burial by bioturbation is considerably more muted or even reversed at higher bioturbation intensities and depths (Fig. 2 and Supplementary Figures 4–6). Similarly, we find that CFA burial does not have a simple, linear relationship with organic matter flux or bottom-water oxygen levels (Fig. 2 and Supplementary Figures 4–6). Changes in bioturbation, organic matter flux, and bottom-water oxygen levels shape multiple factors controlling the saturation state of CFA, which can lead to nonlinear changes in P burial efficiency. For example, higher organic matter fluxes can promote CFA burial by providing more dissolved inorganic phosphate[17]. However, increased organic matter fluxes can also dampen CFA burial, when there is a sharp drop in the pH of pore waters in the upper portion of the sediment pile. Although our model provides new insights into P burial (Fig. 2), our results, overall, corroborate those of other modeling studies that have documented similar nonlinear behavior of CFA burial in response to variability in environmental factors[17].

Despite significant increases in the complexity of our extended diagenetic model, we find, in agreement with the results of our basic diagenetic model, that marine Ca concentrations markedly alter the efficiency of P burial. Calcium concentrations, by strongly influencing CFA saturation state, have a strong impact on overall P burial relative to other environmental factors (Figs. 2–3 and Supplementary Figures 4–7). Unlike other environmental forcings—such as bioturbation, organic matter flux, and bottom-water oxygen levels—higher seawater Ca concentrations always induced more CFA burial (Fig. 2 and Supplementary Figures 4–6), owing to its direct effect on the saturation state of CFA. Increases in CFA burial at high seawater Ca concentrations mediate decreases in the burial ratio between organic C and reactive P ($C_{org}/P_{reac}$, Fig. 2, and Supplementary Figures 4–5). Given significant secular variability in seawater dissolved Ca concentrations over the last 80 million years as well as, more broadly, throughout the Phanerozoic (between ≤10 and 30 mM over the last ~500 million years)[12–14], shifts in dissolved calcium concentrations are likely to have been a key environmental factor controlling P burial efficiency over this interval.

Bottom-water dissolved inorganic carbon (DIC), as well as dissolved sulfate, magnesium, and phosphate concentrations appear to have relatively little impact on CFA burial (Fig. 3). There are likely two chief reasons why bottom-water DIC and phosphate concentrations do not significantly mediate CFA burial. The first is that the stoichiometric abundance of $CO_3^{2-}$ and $PO_4^{3-}$ in CFA is low, relative to that of $Ca^{2+}$ (ref. [11]). Moreover, porewater DIC and dissolved

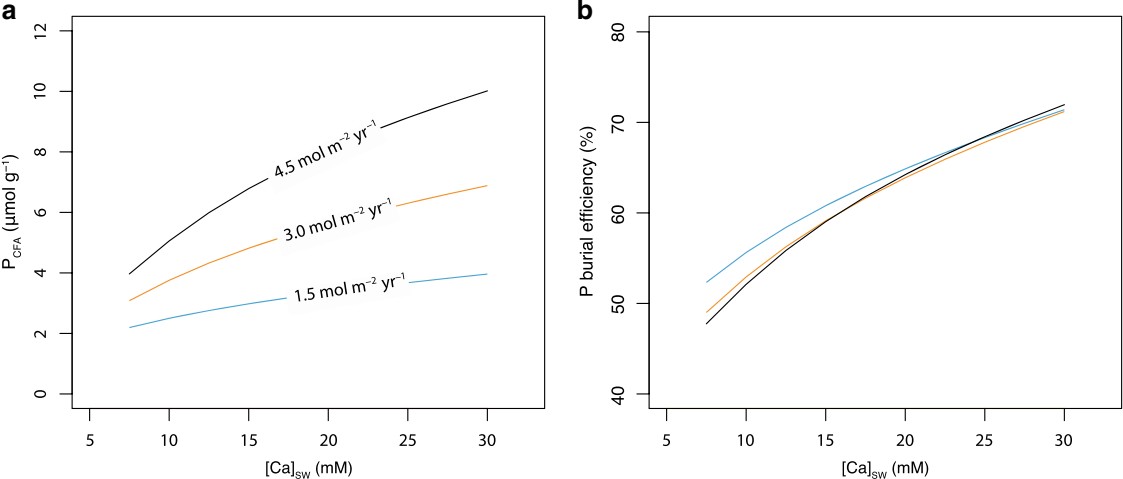

**Fig. 1 Diagenetic model of seawater Ca influence on CFA and P burial.** $P_{CFA}$ represents burial concentration of CFA-associated phosphorus in marine sediments. **a–b** The effects of seawater dissolved calcium concentration ($[Ca]_{SW}$) upon $P_{CFA}$ and P burial efficiency. Labeled fluxes indicate rates of organic matter loading at the sediment–seawater interface.

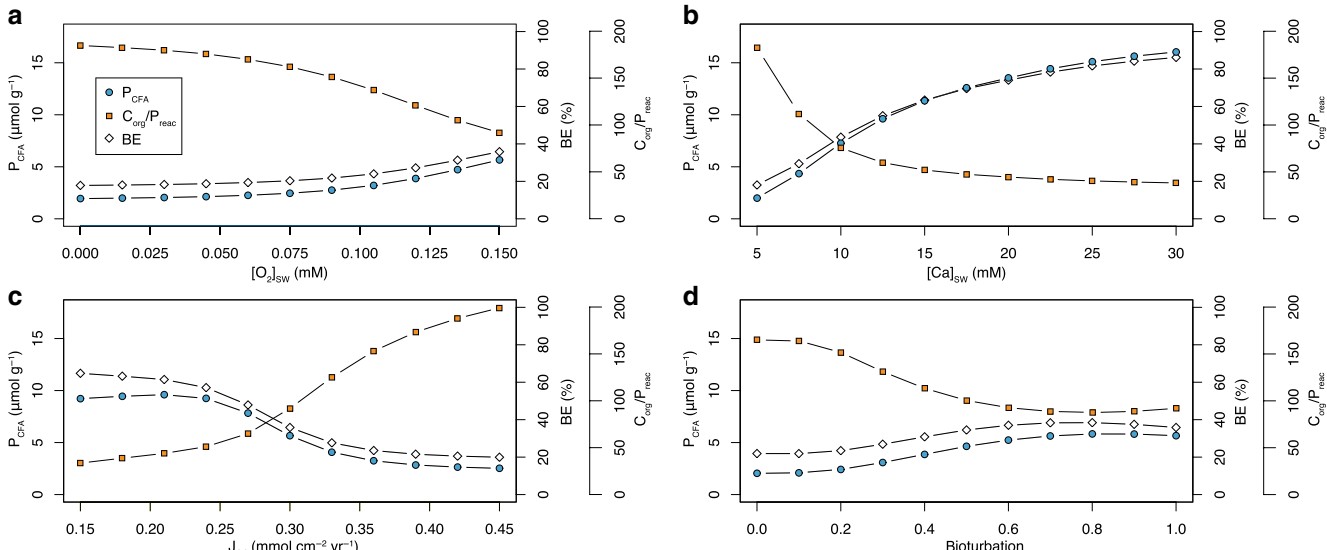

**Fig. 2 Nonlinear effects on CFA, total P burial, and $C_{org}/P_{reac}$.** $P_{CFA}$ (circles) represents burial concentration of CFA-associated phosphorus in marine sediments. BE (diamonds) represents the burial efficiency of reactive phosphorus in marine sediments. $C_{org}/P_{reac}$ represents the burial ratio between organic carbon and reactive P. **a–d** The effects upon $P_{CFA}$, BE, and $C_{org}/P_{reac}$ of bottom-water oxygen concentration **a**, seawater dissolved calcium concentration **b**, flux of organic carbon to the sediment–seawater interface ($J_{OC}$) **c**, and bioturbation **d**. For bioturbation, the four parameters included in the model ($DB_0$, $a_0$, xbt, and xbi) were increased linearly from zero to modern average values, whereas a linear increase of 16% was applied to porosity. Bioturbation is parameterized here as a coupled biodiffusion and bioirrigation term. See Supplementary Tables 1–5 for full list of model parameters.

phosphorus (DIP) are mainly generated by the decomposition of organic matter and/or the reduction of iron oxides[5,8], rather than diffusion from seawater. The Mg concentration of seawater also does not appear to exercise a strong control on CFA burial, likely because Mg is a minor component of CFA. Bottom-water pH also appears to not strongly influence CFA burial (Fig. 3). This is likely because sedimentary and porewater reactions, rather than seawater, are the most important source of the protons within the sediment pile. In sum, we find that bottom-water oxygen, the extent of organic matter loading, and bioturbation play a sizeable role in controlling marine P burial, consistent with previous studies[6,8]. However, our results also show that seawater Ca concentrations have a major role in controlling P burial.

**Empirical evidence for a Ca control on marine P burial.** A Ca control on marine P burial is also supported by P speciation data compiled from Cenozoic deep-sea sediment cores across the Pacific and Atlantic Ocean basins (Fig. 4; see Supplementary Methods; further characteristics of the compiled sites are also shown in Supplementary Table 6). After filtering data from the uppermost sediment pile that are still undergoing active diagenesis (see Supplementary Methods), these deep-sea data suggest relatively constant CFA burial between 80 Ma and 40 Ma and a gradual decrease in CFA burial starting ~40 Ma (Fig. 4). This trend can be seen in both the Pacific and Atlantic sites (Fig. 4). Meanwhile, empirical Mg/Ca data from fluid inclusions, biogenic carbonates, and calcite veins in oceanic crust suggest a gradual decrease in seawater Ca concentrations also began ~40 million

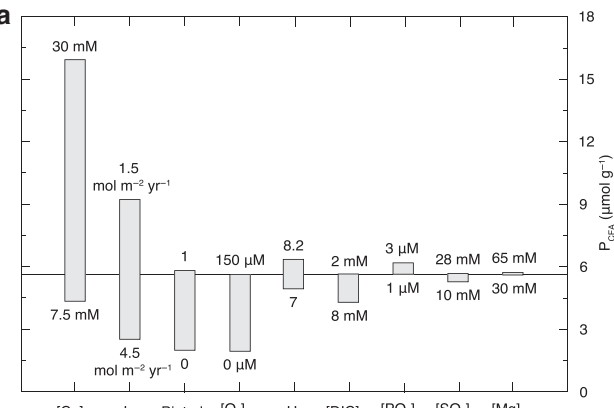
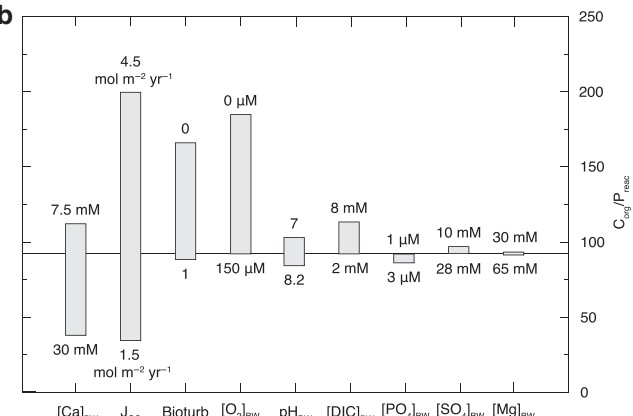

**Fig. 3 Influence of environmental factors on CFA burial. a** The effect of environmental forcings on CFA burial. The *y* axis denotes the concentration of CFA-associated P in sediments. **b** the effect of environmental forcings on the burial ratio between organic carbon and reactive P. The horizontal line represents the value for the shallow marine reference model run (see Supplementary Methods). All parameters were held constant, apart from the individual parameter varied for each sensitivity analysis. Bioturb is a coupled biodiffusion and bioirrigation term and $J_{OC}$ is the flux of organic matter to the sediment-water interface.

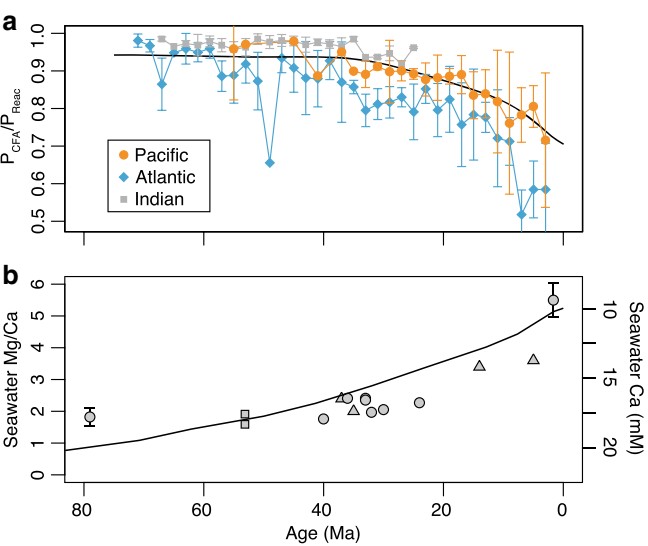

**Fig. 4 Coupling of CFA burial and calcium over 80 million years.**
**a** Changes in the burial ratio of CFA-associated P to total reactive P (a sum of organic P, CFA, and iron-bound P) in deep-sea sediments. Points represent mean values for each 2 myr bin, with the error bars represent one standard deviation (1σ). The black curve represents deep-sea model results from the coupled C-P-O model. See Supplementary Information for discussion of the data compilation. **b** Seawater Mg/Ca ratios recorded in $CaCO_3$ veins of oceanic crust (circles)[14], fluid inclusions (triangles)[12] and echinoderm ossicles (squares)[13]. The curve denotes one of the estimates for seawater Ca concentrations through this interval[12].

years ago (refs. [12–14], Fig. 4). Thus, secular trends in CFA burial correspond well with the timing of changes in seawater Ca concentration. Coupled with the prediction from our model that Ca concentrations will directly affect CFA saturation state and thus CFA burial, this covariation suggests that seawater Ca concentration is, through its influence of CFA formation, a major long-term forcing on global P burial.

**The influence of calcium on atmospheric oxygenation.** In order to explore the influence of variation in marine Ca concentrations on global organic carbon burial and atmospheric $pO_2$, we coupled

our extended diagenetic model to a simple global carbon cycle mass balance model. Specifically, we used the outputs of our extended diagenetic model to parameterize how P burial efficiencies vary as a function of bottom-water oxygen, extent of organic matter loading, marine Ca concentrations, and bioturbation in a global carbon cycle mass balance model (modified from ref. [4], Supplementary Tables 7–13). The primary goal of our global carbon cycle modeling approach was to test to what extent changes in seawater dissolved Ca concentration (as indicated by geologic archives) could have influenced contemporaneous atmospheric $pO_2$ (Fig. 5 and Supplementary Figures 9–14). Full details of the carbon cycle mass balance model (and the full suite of parameters employed) are provided in the Methods and the Supplementary Information. We find that increases in seawater dissolved Ca concentration would, by inducing increased CFA precipitation and decreasing P recycling, lead to decreased seawater P concentration (Fig. 5). More importantly, an increase in CFA precipitation decreases the burial ratio between organic carbon and reactive phosphorus (Fig. 2)—this, at a constant P flux to the oceans, in turn will lead to a drop in atmospheric oxygen levels.

Our model results suggest that Ca could exercise a strong external control on atmospheric $pO_2$ (Fig. 5). For example, an increase in seawater Ca concentration from 10 mM to 20 mM could lead to a >50% decrease in atmospheric $pO_2$, from modern levels 21% to ~10% (Fig. 5). There is a small rebound in the total marine P reservoir after a decrease induced by a shift in $[Ca^{2+}]$ (Fig. 5c), which is owing to the delayed decrease in atmospheric oxygen and thus iron-bound P burial.

Given the large swings in Phanerozoic seawater Ca concentration recorded by geologic archives[12–14,18,19], our results suggest that seawater Ca concentration could be one of the key factors shaping atmospheric $pO_2$ evolution. For example, proxy records indicate low seawater dissolved Ca concentrations during the Carboniferous–Permian[12,20], which would have inhibited CFA formation and driven the development of the high atmospheric $pO_2$ that, on the basis of both geochemical and paleontological archives[21–25], has been previously suggested to be characteristic of this time (interval 3 of Supplementary Figure 12). Fluid-inclusion data and carbonate mineralogical records are also commonly interpreted to indicate an increase in seawater dissolved Ca concentrations occurred during the early Cambrian[12,26–28], coincident with what has been previously

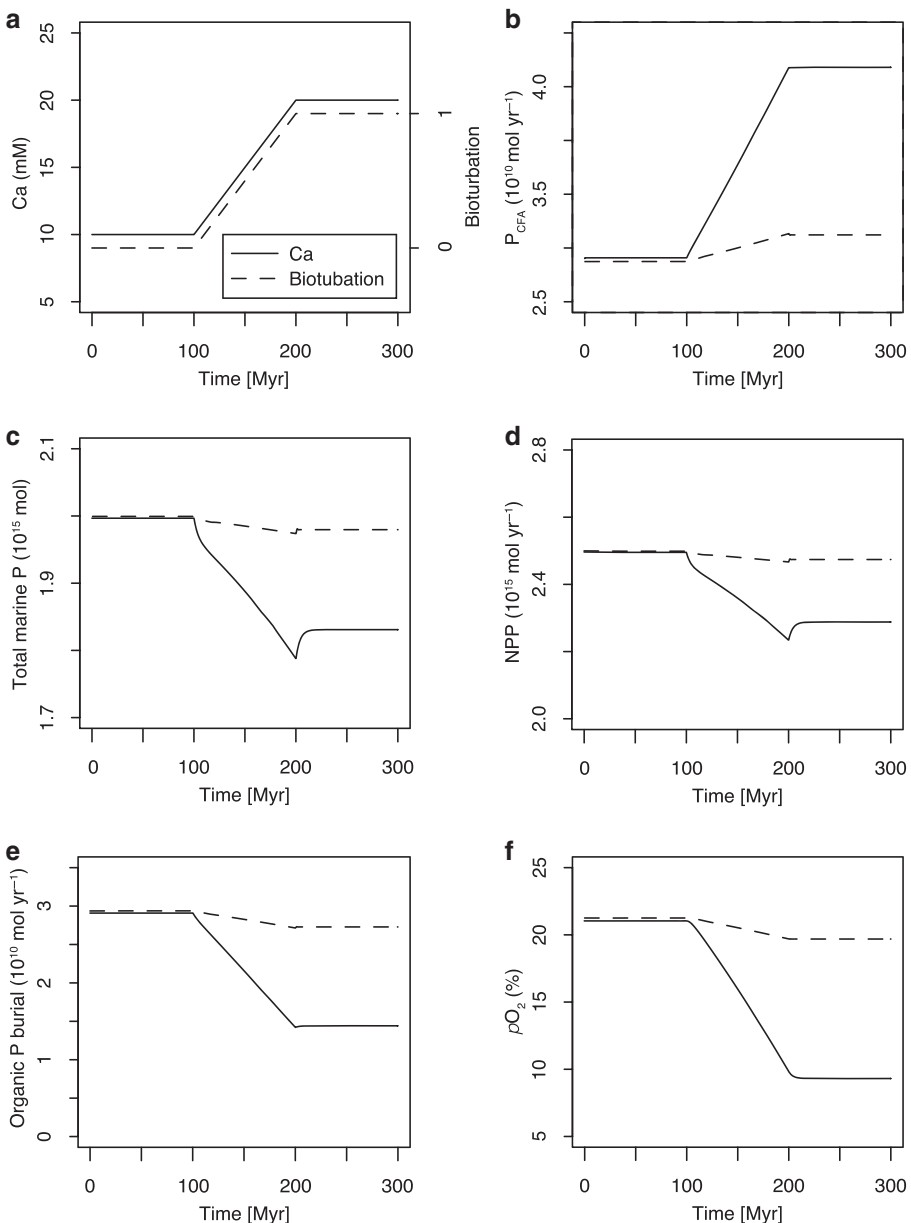

**Fig. 5 Atmospheric oxygen levels respond to changes in calcium or bioturbation. a** Model input of seawater dissolved Ca concentration or bioturbation (0 represents no bioturbation, whereas 1 represents the reach of modern bioturbation). **b–f** Model outputs of variations in P burial associated with CFA formation, total marine P (a sum of dissolved inorganic phosphate, dissolved organic phosphorus and soluble particulate inorganic phosphorus), net primary productivity (NPP), organic P burial flux and atmospheric oxygen levels. $pO_2$ shown in the figure is the actual percent by volume.

suggested to have been an interval of ocean deoxygenation (e.g., refs. [6,23,29]).

Our modeling results indicate that seawater Ca concentration is likely a major driver of total P burial, such that, over Earth's history, high seawater dissolved Ca concentrations should correspond to significant increases in P burial efficiency and relatively low atmospheric $pO_2$. A compilation of deep-sea P speciation data from the past 80 million years provides empirical support for the effect of seawater dissolved Ca concentrations on P burial and bolster the case that this Ca-driven feedback strongly impacts the global ocean-atmosphere system. These results provide a new view of the processes linking marine elemental cycles, tectonics and atmospheric $pO_2$, and suggest that the major ion composition of seawater has been an important driver of biospheric change and atmospheric evolution throughout Earth's history.

## Methods

**Basic 1D diagenetic model.** The basic reactive-transport diagenetic model we developed includes six components, including two solid-phase species: organic matter and CFA; and four solute components: $[Ca^{2+}]$, $[DIC]$, $[DIP]$, and $[F^-]$. The mass balance functions for this basic model are[30,31]:

$$\frac{\partial C_l}{\partial t} = \frac{1}{\phi}\frac{\partial}{\partial x}\left(\phi D \frac{\partial C_l}{\partial x}\right) - \frac{1}{\phi}\frac{\partial}{\partial x}(\phi v C_l) + \sum R_l \quad (1)$$

$$\frac{\partial C_s}{\partial t} = -\frac{1}{1-\phi}\frac{\partial}{\partial x}((1-\phi)\omega C_s) + \sum R_s \quad (2)$$

where $C_l$ is the concentration of solute, $C_s$ is the concentration of solid, $\phi$ is porosity, D is molecular diffusion, $v$ is the solute advection rate, $\omega$ is solid advection rate, and $R_s$ and $R_l$ are, respectively, the solid and solute reaction rates. The effect of molecular diffusion is calibrated to tortuosity through $D = \frac{D_m}{1-2\ln(\phi)}$, where $D_m$ is the intensity of molecular diffusion. The effect of compaction on the solute and solid advection rates was also considered using the method described in ref. [30]. The depth dependence of porosity was represented by $\phi(x) = \phi_\infty + (\phi_0 - \phi_\infty)\exp(-x/\lambda)$, where $\lambda$ is the porosity attenuation length, and $\phi_0$ and $\phi_\infty$ represent

porosity at the sediment–seawater interface and at depth, respectively. The boundary conditions for these species and the above model parameters were also used for the extended, multicomponent version of the diagenetic model (described in more detail below; Supplementary Tables 1 and 5). The porewater profiles of pH, $[Mg^{2+}]$, and $[Na^+]$ were, for this basic iteration of the diagenetic model, fixed at seawater levels. The Redfield ratio of 106:1 was used as the C/P ratio of the organic matter. There are only two reactions in this basic model, the decomposition of organic matter and the formation of CFA. A reactive continuum-type model[32] was used to describe the decomposition of organic matter in this basic model iteration.

CFA is the largest P sink in the global marine P cycle[3]. In this study, we use the stoichiometry of a carbonate-rich francolite[11,33]—$Ca_{9.54}Na_{0.33}Mg_{0.13}(PO_4)_{4.8}$ $(CO_3)_{1.2}F_{2.48}$—although the stoichiometry of CFA can be variable. Previous studies have simplified the precipitation rate of CFA by presuming a linear relationship with excess dissolved phosphate (relative to an assumed equilibrium phosphate concentration, refs. [5,8,10]). However, experimental studies have shown that the rates of the crystal growth of calcium hydroxylapatite and calcium fluorapatite are functions of saturation state[15,16]. Thus, in this study, we more completely parameterize the kinetics of CFA formation by assuming that the formation rate of CFA is characterized by a first-order relationship with its saturation state. When the saturation state of CFA ($\Omega_{CFA}$) is higher than 1, the precipitation rate of CFA can be expressed as:

$$R_{CFA} = k_{CFA}(\Omega_{CFA} - 1) \qquad (3)$$

where $k_{CFA}$ is a kinetic constant for CFA, obtained by fitting to FOAM porewater and sedimentary CFA profiles. $\Omega_{CFA}$ is expressed as:

$$\Omega_{CFA} = \frac{([Ca^{2+}] \cdot rCa)^{9.54} \cdot ([Na^+] \cdot rNa)^{0.33} \cdot ([Mg^{2+}] \cdot rMg)^{0.13} \cdot ([PO_4^{3-}] \cdot rPO_4)^{4.8} \cdot ([CO_3^{2-}] \cdot rCO_3)^{1.2} \cdot ([F^-] \cdot rF)^{2.48}}{K_{spCFA}} \qquad (4)$$

where $rCa$, $rNa$, $rMg$, $rPO_4$, $rCO_3$, and $rF$ represent the activity coefficients of each ion. $K_{spCFA}$ is the equilibrium constant of CFA, which was found to be a function of carbonate activity[11]:

$$\log(K_{spCFA}) = -83.231 + 2.3307 \cdot \log([CO_3^{2-}] \cdot rCO_3) \qquad (5)$$

In this parametrization, carbonate ion activity influences the saturation state of CFA through its effect on both IAP and Ksp. We have also used the same stoichiometry of CFA and a fixed $K_{spCFA}$ ($10^{-99.7}$) without a relationship with carbonate activity in a series of runs (Supplementary Figures 5, 6, and 10) and found that this change only has a subtle influence on the model results (Figs. 2 and 5, and Supplementary Figures 5 and 10). It is also possible that stoichiometric abundance of $CO_3^{2-}$ in CFA correlates with $[CO_3^{2-}]$ of porewater[11]. However, the relationship between the solubility of CFA, the stoichiometric abundance of $CO_3^{2-}$ in CFA and $[CO_3^{2-}]$ of porewater are not fully understood[11]. These uncertainties may influence the relationship between $[CO_3^{2-}]$ of porewater and CFA formation, which is currently a negative correlation (Fig. 3). Further, these will not influence our main conclusions as they do not significantly influence the relationship between $[Ca^{2+}]$ and the saturation of CFA ($\Omega_{CFA}$). Following the approach of previous studies[8,10], the dissolution of CFA under $\Omega_{CFA} < 1$ is not included, as CFA is highly insoluble in marine sediments.

**Extended 1D multicomponent diagenetic model.** Building from previous efforts[8,34–36], we also built an extended 1D multicomponent model incorporating the biogeochemical cycles of C, N, P, S, Fe, and Mn to more fully simulate the diagenetic P cycle in marine sediments (SEDCHEM). The model includes 15 solutes and 21 solids (Supplementary Table 1). A biodiffusion term was used to describe the mixing of sediment particles. A non-local function[37,38] was applied to describe the influence of bioirrigation on the exchange of solutes near the sediment–seawater interface. Combining the molecular diffusion, advection, bio-turbation, and reaction terms, the mass balance of solutes and solids can be generalized to the following functions[30,31]:

$$\frac{\partial C_l}{\partial t} = \frac{1}{\phi}\frac{\partial}{\partial x}\left(\phi D \frac{\partial C_l}{\partial x}\right) - \frac{1}{\phi}\frac{\partial}{\partial x}(\phi v C_l) + a(C_{l0} - C_l) + \sum R_l \qquad (6)$$

$$\frac{\partial C_s}{\partial t} = \frac{1}{1-\phi}\frac{\partial}{\partial x}\left((1-\phi)D_B\frac{\partial C_s}{\partial x}\right) - \frac{1}{1-\phi}\frac{\partial}{\partial x}((1-\phi)\omega C_s) + \sum R_s \qquad (7)$$

where $D_B$ is the intensity of biodiffusion, $a$ is the coefficient of bioirrigation and $C_{l0}$ is the solute concentration in open burrows, which is assumed to be equivalent to the solute concentration of the overlying water column. The attenuations in the intensities of biodiffusion and bioirrigation at depth are described by $D_B(x) = D_{B0} \cdot \exp\left(-\left(\frac{x}{xbt}\right)^2\right)$ and $a(x) = r \cdot a_0 \cdot \exp\left(-\frac{x}{xbi}\right)$, respectively. $D_{B0}$ and $a_0$ are the biodiffusion and bioirrigation intensities at the sediment–seawater interface, $xbt$ and $xbi$ are the respective attenuation coefficients and $r$ is a correction for the irrigation of $Fe^{2+}$ and $Mn^{2+}$ (refs. [35,39]). Model components and boundary conditions are shown in Supplementary Table 1. Model reactions, reaction rate laws and parameters can be found in Supplementary Tables 2–5. The following three sections delineate treatment of the diagenetic P cycle, pH, and adsorption in the

extended 1D multicomponent diagenetic model. See Supplementary Text for model solution and model applications.

**Diagenetic phosphorus cycle.** Parameterizations of the diagenetic P cycle explored in this study share many features with previously published model exercises (e.g., refs. [8,10,36]). Three solid P species—organic phosphorus (orgP), iron-bound phosphorus ($P_{Fe}$), and CFA—as well as dissolved inorganic phosphate ($\Sigma PO_4^{3-}$) are included in the extended version of our diagenetic model. In contrast to previous studies, however, our model also includes a more intrinsic reaction rate law for CFA formation, which involves all the major components of CFA and parameterizes the effect of pH on P speciation and burial. To simulate pH, our model also includes proper descriptions of adsorption, the diagenetic Fe cycle, and authigenic carbonate precipitation and dissolution.

Decomposition of organic matter drives diagenetic reactions. The chemical formula of organic matter can be simplified as $CN_{rN}P_{rP}$, where $rN$ and $rP$ are the molar ratios of organic nitrogen and organic phosphorus relative to organic carbon. The major pathways for the decomposition of organic matter, including aerobic respiration, nitrate reduction, Mn reduction, Fe reduction, sulfate reduction, and methanogenesis are included in the model. As the sequence (and thus spatial distribution) of these pathways is dictated by their respective energy yields[40], a Monod scheme was used to describe the operation of these reactions[8,10,36].

For the mineralization of orgC in the extended model, we used a multi-G model[41] developed from a reactive continuum-type model[32]. The advantage of this model is that it not only can represent the reactive continuum of organic matter, but also is suitable to apply to the bioturbated zone of the sediment pile[41]. We divided organic matter into 12 G in this study. Although each G has its own first-order rate constant, their fractions in total organic matter are determined by a gamma distribution[41]. The rate constant for each $G_i$ is

$$k_i = \begin{cases} 1, i = 1 \\ 10^{1.5-i}, i = 2 \ to \ 11 \\ 10^{-10}, i = 12 \end{cases} \qquad (8)$$

while the fraction of each $G_i$ is

$$f_i = \begin{cases} \frac{\int_0^\infty x^{v-1} \cdot e^{-x} dx - \int_0^a x^{v-1} \cdot e^{-x} dx}{\int_0^\infty x^{v-1} \cdot e^{-x} dx}, i = 1 \\ \frac{\int_0^{a \cdot 10^{2-i}} x^{v-1} \cdot e^{-x} dx - \int_0^{a \cdot 10^{1-i}} x^{v-1} \cdot e^{-x} dx}{\int_0^\infty x^{v-1} \cdot e^{-x} dx}, i = 2 \ to \ 11 \\ \frac{\int_0^{a \cdot 10^{-10}} x^{v-1} \cdot e^{-x} dx}{\int_0^\infty x^{v-1} \cdot e^{-x} dx}, i = 12 \end{cases} \qquad (9)$$

where $a$ is the average lifetime of more reactive orgC and $v$ is the shape of orgC distribution[32].

The C/P ratio of particulate organic matter typically increases with depth in the sediment pile. Following previous methods[36,42], we assume that this variation is generated by different C/P ratios in different organic components, with less reactive (more refractory) organic matter having higher C/P ratios. Thus, we further divided the 12 G fractions into two pools ($\alpha$ and $\beta$) with different C/P ratios. With this procedure, it is possible to reproduce empirical sedimentary profiles of organic phosphorus and dissolved phosphate (Supplementary Figure 2). The $\alpha$ pool includes the first 2 G fractions ($G_1$ and $G_2$), whereas the $\beta$ pool includes the remaining 10 G fractions ($G_3$–$G_{12}$). The C/P ratios of the two pools are shown in Supplementary Table 5.

Five iron phases are included in the modeled flux to the sediment–seawater interface: (1) highly reactive iron hydroxides ($Fe(OH)_3^\alpha$); (2) less reactive iron hydroxides ($Fe(OH)_3^\beta$); (3) unreactive iron hydroxides ($Fe(OH)_3^\gamma$); (4) magnetite ($Fe_3O_4$); and (5) biotite (Biot). Demarcation of iron hydroxides by reactivity is similar to the treatment employed by previous models[8,36]. The rate law for magnetite dissolution is reasonably well established[43]. Iron may also occur as other silicate-bound phases, but we use biotite as a representation of silicate iron (see ref. [44]). At FOAM, biotite dissolution appears to be closely linked to porewater pH. As for previous models[8], iron-bound phosphorus is assumed to be associated with iron hydroxides, and the P/Fe molar ratio is described using a constant $\gamma$ and a variable $\theta$ (Supplementary Tables 1–5). Thus, the precipitation and dissolution of iron hydroxides are accompanied by, respectively, the scavenging and release of dissolved phosphate (Supplementary Table 2).

A major difference of this study from previous studies[5,8,10] is that we have parameterized the formation rate of CFA using its saturation state (see above), which is consistent with experimental results[15,16]. A full description of this method can be found above, as well as in the Supplementary Text and Supplementary Table 3.

We have also included the iron phosphate mineral vivianite in the extended model. Although it has not been extensively documented in marine sediments, it is commonly found in restricted settings[45]. Following ref. [45], we have used the Michaelis–Menten kinetics for dissolved phosphate and iron in marine porewater to describe the formation of vivianite. Details of the formulation and parameters for vivianite can be found in Supplementary Tables 3 and 5.

**Diagenetic pH simulation**. We used a method similar to that of Hoffman et al.[46] to simulate pH variation during diagenetic processes. Instead of total alkalinity, we used total proton balance (TP) as an equilibrium-invariant and implicit differential variable (whose differentials $\frac{dy}{dt}$ are not treated as a main function and are only used to calculate the $\frac{dy}{dt}$ of the differential variable) to model pH. Although the results of the two methods are the same, the new method is more straightforward and less-demanding computationally. We define the total proton balance as the sum of those ions that can undergo proton exchange at the modeled pH range (~6–9), which can be written as:

$$TP = 2CO_2 + HCO_3^- + NH_4^+ + 2H_2PO_4^- + HPO_4^{2-} + H_2S + H^+ \quad (10)$$

As the first dissociation constant of phosphoric acid is high and the second dissociation constant of hydrogen sulfide is very low, we do not include $H_3PO_4$ and $HS^-$ in the formulation of TP. Following Hoffman et al.[46], we treat TP as an implicit differential variable in the model, which removes the need to solve the algebraic equation numerically. The derivative of the proton concentration is:

$$\frac{dH}{dt} = \left( \frac{dTP}{dt} - \sum_j \frac{\partial TP}{\partial [S_j]} \frac{d[S_j]}{dt} \right) / \frac{\partial TP}{\partial H} \quad (11)$$

where $[S_j]$ is the total concentration of the dissociation systems ($\sum CO_2$, $\sum NH_3$, $\sum H_3PO_4$, and $\sum H_2S$). A derivation of $\frac{dH}{dt}$ and the formulation of $\frac{\partial TP}{\partial [S_j]}$ and $\frac{\partial TP}{\partial H}$ can be found in the Supplementary Text. In the extended model, the proton mass balance is calculated using Eq. 11, and the $\frac{dTP}{dt}$ term is the sum of the mass balance functions of all its species calculated using Eq. 6.

**Diagenetic adsorption simulation**. Adsorption within marine sediments often involves exchange between protons and ions such as $Fe^{2+}$ and $Mn^{2+}$, which is important in modeling pH variation during diagenetic processes. Adsorption is treated here as a reversible linear equilibrium process. For instance, for the adsorption of $Fe^{2+}$, $K_{Fe}$ can be defined as the amount of iron in the adsorbed phase relative to $Fe^{2+}$ in the solute, thus

$$A_{Fe} = FK_{Fe}Fe^{2+} \quad (12)$$

where $A_{Fe}$ is the concentration of adsorbed solid-phase iron, and $F = (1 - \phi)/\phi$. In the model, $Fe^{2+}$ is treated as a differential variable, whereas $A_{Fe}$ is treated as an algebraic variable, with the assumption of instantaneous equilibrium between them. As $Fe^{2+}$ is influenced by both the reaction and transport of $Fe^{2+}$ and $A_{Fe}$, the derivative of $Fe^{2+}$ is

$$\frac{dFe^{2+}}{dt} = \frac{1}{1 + K_{Fe}} RT_{Fe^{2+}} + \frac{F}{1 + K_{Fe}} RT_{A_{Fe}} \quad (13)$$

where $RT_{Fe^{2+}}$ and $RT_{A_{Fe}}$ are the sum of the reaction and transport terms (excluding adsorption) of $Fe^{2+}$ and $A_{Fe}$, respectively, which are calculated from Eqs. 6 and 7. The transfer rate ($RA_H$) of protons during the adsorption process is

$$RA_H = \frac{K_{Fe}}{1 + K_{Fe}} RT_{Fe^{2+}} - \frac{F}{1 + K_{Fe}} RT_{A_{Fe}} \quad (14)$$

Derivations of Eqs. 13 and 14 can be found in the Supplementary Text. The adsorption of $NH_4^+$ was not included in the model as it does not influence pathways of P cycling. However, the C/N ratios of organic matter were tuned to fit the FOAM porewater profile (Supplementary Figure 2). As the model can, without parameterization of $Mn^{2+}$ adsorption, accurately reproduce the FOAM dissolved $Mn^{2+}$ profile (Supplementary Figure 2), the adsorption of $Mn^{2+}$ was also not included.

**Coupled carbon–phosphorus–oxygen cycle model**. Our coupled carbon–phosphorus–oxygen cycle model integrates our newly developed diagenetic model with a global biogeochemical cycling model modified from the carbon–phosphorus cycle model of Van Cappellen and Ingall[4,47]. The purpose of this coupled diagenetic-global biogeochemical cycling modeling exercise is to illustrate the relative importance of various environmental and biotic factors in controlling carbon and phosphorus cycling and atmospheric $pO_2$ levels. Flux equations, reservoir equations, and parameters are similar to those presented in Van Cappellen and Ingall[4,47]. Detailed description of the reservoirs, fluxes, and parameters used in the model (and values for each of these) are listed in Supplementary Tables 9–12.

In contrast to the approach of Van Cappellen and Ingall[4,47], in our coupled model the marine sedimentary burial of calcium-bound phosphate (F58) is calculated from our diagenetic model (interpolated between time slices of 1 million years). Rates of organic carbon burial (which are, in turn, influenced by organic carbon remineralization) and organic carbon weathering are, in our model, influenced by atmospheric $O_2$ level, following the parameterization of the COPSE model[48]. In particular, we have also, for these runs, parameterized the organic C/P ratio as a function of seawater oxygen level in our diagenetic model. This is achieved by parameterizing two factors rP1 and rP2 using the

following functions:

$$rP1 = \begin{cases} \frac{1}{106}, & [O_2]_{BW} > 150\,M \\ \frac{1}{106} * \frac{[O_2]_{BW}}{150} + \frac{2.4}{106} * \left(1 - \frac{[O_2]_{BW}}{150}\right), & [O_2]_{BW} < 150\,M \end{cases} \quad (15)$$

$$rP2 = \begin{cases} \frac{1}{106}, & [O_2]_{BW} > 150\,M \\ \frac{1}{106} * \frac{[O_2]_{BW}}{150} + \frac{1}{500} * \left(1 - \frac{[O_2]_{BW}}{150}\right), & [O_2]_{BW} < 150\,M \end{cases} \quad (16)$$

To couple the diagenetic model and the carbon–phosphorus–oxygen cycle model, we built high-resolution look-up tables for CFA burial during the Phanerozoic, using the results of our diagenetic model. We carried out a series of model runs with different bottom-water oxygen concentrations and organic carbon fluxes to the sediment–seawater interface to build look-up tables for the deep sea and shallow oceans, respectively, which were then used to force P burial at each time step. More discussion of the coupled carbon–phosphorus–oxygen cycle model can be found in the Supplementary Text.

## Data availability
The authors declare that the data supporting the findings of this study are available within the paper and its supplementary information files.

## Code availability
Codes are available as supplementary files.

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

## Acknowledgements
We are extremely grateful to Edward Bolton for his help in building the models.

## Author contributions
M.Z. and N.P. wrote the paper with contributions from L.G.T., C.T.R., and S.Z. All authors were involved in the design of the study and the design of the utilized models. S.Z. and M.Z. carried out the carbon cycle modeling. M.Z. carried out the diagenetic modeling. N.P. supervised the project.

## Competing interests
The authors declare no competing interests.
