## [Peer Review File · Nature Communications]

Reviewers' comments:

Reviewer #1 (Remarks to the Author):

This paper is well written and introduces the idea that the burial of calcium phosphate minerals is primarily controlled by ocean calcium concentrations, and that therefore the Ca content of ancient seawater is an unrecognized and powerful control on atmospheric O₂ evolution. This is potentially an important finding, but I have some concerns with how the idea is currently developed and reported:

I agree with the authors that "shifts in dissolved calcium concentrations are likely to have been a key environmental factor controlling P burial efficiency", but I do not think they have adequately shown that "seawater Ca abundance—rather than bioturbation or redox state—is the strongest driver of total P burial". Figure 3 shows the relative importance of different factors but only shows CFA burial, therefore omits half or more of the total P burial flux. In figure 2 the authors do show total P burial efficiency, but here the relative burial efficiency is strongly affected by all of the parameters tested e.g. BE doubles over the Phanerozoic range of [Ca] but also appears to change by this factor or more over the other ranges shown? So currently the conclusion that [Ca] is the dominant control on overall P burial has not been satisfactorily shown.

Secondly, the authors write that the goal of the final section and box model is to "explore whether predicted shifts in P cycling would drive significant changes in surface O₂ levels." rather than to reconstruct atmospheric O₂. But if this is the case then it should be presented in a different way, showing the response of O₂ to varying other important parameters and processes (weathering and degassing fluxes, bioturbation etc.). Currently the analysis makes a Phanerozoic O₂ estimation, but does so based on only a handful of processes and inadequate Ca data, so it just doesn't really make sense.

Finally, the paper uses some overstated phrases, such as "This is in striking contrast to the previous understanding on the environmental forcings of P burial." This should really be toned down. The fact that [Ca] promotes the formation of Ca minerals is neither striking nor in contrast to traditional thinking!

Reviewer #2 (Remarks to the Author):

The authors show convincingly that seawater Ca has a strong effect on CFA formation and global P burial in marine sediments. This is an important conclusion that justifies a publication of their manuscript in Nature Communications.

Even though the diagenetic model presented by the authors is much more advanced than previous models of P diagenesis, the model has some limitations that should be acknowledged by the authors. Most importantly, their diagenetic model does not consider the effect of oxygen on organic P degradation and organic C/P burial ratios in marine sediments. Organic C/P ratios in sediments deposited under permanently anoxic bottom water conditions can be extremely high (up to 1000) which indicates that bottom water oxygen has a strong effect on P burial. This important observation is not reproduced by the model. Therefore, the authors cannot claim that the Ca effect is stronger than the oxygen effect and that Ca is more important than all other factors regulating P burial (e.g. line 20). They can claim that Ca is important but should not overstate i.e. write that it is more important than anything else.

Moreover, the authors introduce a new approach to calculate porewater pH. This model routine is important because it governs the carbonate ion concentration level in porewater that has a significant

effect on CFA formation. The description of this routine is, however, difficult to follow and I am not convinced that the pH calculation fully considers the proton turnover induced by the various reactions that take place in the model (Tab. S2). I do not understand how the proton concentration (H) is treated in the model. H appears as one of the species in Tab. S1. Does this mean that the transport-reaction Eq. 6 is solved for H or is Eq. 14 used to calculate H? If Eq. 14 is applied then the proton turnover induced by the various reactions in Tab. S2 may not be fully considered in the model. The authors should better explain how H is calculated and how the proton turnover is considered in the pH model.

I added a number of additional comments to the supporting information file attached to this review.

Reviewer #3 (Remarks to the Author):

Review of Zhao et al., for Nat. Comm.

Zhao et al. explore an interesting idea in this contribution that changes in seawater Ca^{2+} concentrations over geologic time could have affected the regeneration of P from sediments. Their idea is straightforward and sound — Phosphorous is trapped in sediments either as organic matter, bound to iron, or precipitated as apatite (carbonate fluorapatite [CFA] dominantly). Oversaturation for CFA occurs in sediments when the rate of phosphorus release from organic carbon degradation outpaces the diffusional loss of P to the subsurface allowing CFA to become oversaturated and precipitate. Seawater Ca^{2+} concentrations have varied by a factor (we think based on fluid inclusion data) of about 2-3 over the Phanerozoic. Thus, it is reasonable to explore the idea that elevated Ca^{2+} levels in past oceans could have served to make it easier to precipitate CFA in the sediments for a given phosphorus level in the oceans. In terms of the most basic idea of the paper, it is sound and interesting and worthy of exploration and will have useful impact on experts in the field.

However, how the authors go about exploring this idea in a quantitative sense I could not follow nor evaluate in the main text. Three separate models are developed. Two are sedimentary reaction-diffusion equations and these are then connected to a global carbon cycle model to explore the results over the past 500 million years. This involves 96 separate differential equations (many of which are coupled) and requires assumptions about rate constants and the like — note I just counted up the number of equations in the SI, some may be redundant, but regardless, the number is high. However, none of the equations used are presented in the main text nor are they discussed. None of the assumptions made are discussed. How various rate constants were chosen is not discussed in the main text. Rather all of this is placed in the supplementary information. As a result, in this contribution, the supplementary information, rather than adding extra information to provide a curious reader additional context for the main manuscript, is required reading and must be read along with the main manuscript both to understand exactly what it is the authors have done, and to be able to evaluate the results. In other words, in my reading, the main manuscript does not stand on its own. Based on this, I cannot recommend the paper as written to Nature Communications as a reader. This is not a criticism of the paper's substance (which I like, though I found many aspects confusing due to the large amount of info in the SI). Instead, I would encourage the authors to submit to a journal like AJS or GCA in which their models are fully explained and contextualized, assumptions made clear, and model space explored. In other words, the manuscript, as it stands, does not stand on its own and requires careful co-reading of the SI to follow and evaluate the model outputs.

Specific comments:

(1) Something that was not clear to me in reading the paper is how the calculations were done. Specifically, is the dissolved phosphorus content of the ocean held constant as Ca^{2+} is varied for

example? Does raising Ca lower this (given that there is less P removed from the sediments I assume it must). Given that the system, I assume, reaches a steady state in which P inputs equal P outputs, does the regeneration really matter if what goes in to the ocean goes out on geologic timescales? I.e., when a perturbation is forced (a change in Ca²⁺) how long does the system take to relax such that inputs equal outputs. Figure 4 appears to indicate, at least to me, an immediate response. Figure 3 shows some of this variation, but I would like to see the relationship between Ca vs PO₄ in seawater for steady state solutions.

(2) In figure 4, we see PCFA/Preactive plotted. How does P reactive change with time vs. P total? And how do C/P ratios change. I found this comparison confusing as without knowing how the denominator (P reactive) is changing with time, it is unclear to me if there is actually less P available back released to the ocean, or not. Something else that concerned me about this plot is that the PCFA/Preactive has the highest uncertainties for the modern samples, and the lowest uncertainties for the ancient samples. Is there an aliasing going on here that we should worry about?

(3) Figure 4b is important as it shows PCFA burial changing with time that inversely correlates with pO₂ (indicating a coupling). But how does total P burial change with time in the model? I'm assuming this is constant (or at least that P weathering equals P burial on ~million year timescales). Does the P flux to oceans change? Is the C/P ratio of sediments changing? The graph implies that PCFA is important. But does it really matter what PCFA fluxes are vs. total fluxes? Alternatively, I would think the P regeneration rate is really what matters (i.e., whether you get P out of sediments, but leave Corg behind). This will impact C/P ratios.

Reviewers' comments:

Reviewer #1 (Remarks to the Author):

This paper is well written and introduces the idea that the burial of calcium phosphate minerals is primarily controlled by ocean calcium concentrations, and that therefore the Ca content of ancient seawater is an unrecognized and powerful control on atmospheric O₂ evolution. This is potentially an important finding, but I have some concerns with how the idea is currently developed and reported:

I agree with the authors that “shifts in dissolved calcium concentrations are likely to have been a key environmental factor controlling P burial efficiency”, but I do not think they have adequately shown that “seawater Ca abundance—rather than bioturbation or redox state—is the strongest driver of total P burial”. Figure 3 shows the relative importance of different factors but only shows CFA burial, therefore omits half or more of the total P burial flux. In figure 2 the authors do show total P burial efficiency, but here the relative burial efficiency is strongly affected by all of the parameters tested e.g. BE doubles over the Phanerozoic range of [Ca] but also appears to change by this factor or more over the other ranges shown? So currently the conclusion that [Ca] is the dominant control on overall P burial has not been satisfactorily shown.

Response: This is a fair critique. We (as a community) are still moving forward our understanding of how water-column oxygen levels and bioturbation—among other factors—shape P burial (with several exciting recent papers on the evolution of P cycling). We have removed all statements to the effect that Ca levels are the most important factor (as we concede that this conclusion was derived only from the parameter space we explored). We have modified the text to make it clear that our central conclusion is solely **that marine Ca concentrations play a critical and previously underappreciated role in controlling marine P burial**. Therefore, secular shifts in the marine Ca cycle must be taken into account when trying to reconstruct the factors controlling past and present extents of primary productivity, organic carbon burial, and atmospheric oxygen levels. We have also made it clear that our global redox mass balance model is not designed to give the most accurate possible reconstruction of Phanerozoic pO_2 evolution—it is simply designed to demonstrate that, with accepted trends in marine Ca concentrations, the evolution of the Ca cycle will drive significant shifts in atmospheric oxygen levels.

Secondly, the authors write that the goal of the final section and box model is to “explore whether predicted shifts in P cycling would drive significant changes in surface O₂ levels.” rather than to reconstruct atmospheric O₂. But if this is the case then it should be presented in a different way, showing the response of O₂ to varying other important parameters and processes

(weathering and degassing fluxes, bioturbation etc.). Currently the analysis makes a Phanerozoic O₂ estimation, but does so based on only a handful of processes and inadequate Ca data, so it just doesn't really make sense.

Response: We have applied shifts in marine Ca concentrations documented from available geologic archives (and previously proposed global oxygen cycle parameterizations and values) to see its influence on atmosphere oxygen (Fig. 5). The results show that shifts in P cycling associated with these documented changes in seawater Ca concentration will significantly alter atmospheric oxygen levels. An increase of seawater Ca concentration will, by mediating increased CFA precipitation, cause marine dissolved P (DIP) levels to decrease. This decrease will, in turn, induce a drop in atmospheric oxygen levels. These points are justifiable in spite of uncertainties associated with current estimates of past marine Ca concentrations. Given that we are using the structure and parameters of a previously developed (albeit slightly modified) oxygen cycle model rather than introducing a new model, we have not discussed the sensitivity a range of parameters (e.g., bioturbation and uplift) in detail. However, we have now included additional sensitivity analyses in the Fig.5 and Supplementary Figure 4.

Finally, the paper uses some overstated phrases, such as “This is in striking contrast to the previous understanding on the environmental forcings of P burial.” This should really be toned down. The fact that [Ca] promotes the formation of Ca minerals is neither striking nor in contrast to traditional thinking.

Response: We have deleted this sentence. And we agree that this is unsurprising—the mechanism linking Ca concentrations to P burial (and thus O₂ levels) is very straightforward. However, to the best of our knowledge, shifts in marine Ca levels have not been previously discussed or incorporated into attempts to model the long-term P cycle.

Reviewer #2 (Remarks to the Author):

The authors show convincingly that seawater Ca has a strong effect on CFA formation and global P burial in marine sediments. This is an important conclusion that justifies a publication of their manuscript in Nature Communications.

Even though the diagenetic model presented by the authors is much more advanced than previous models of P diagenesis, the model has some limitations that should be acknowledged by the authors. Most importantly, their diagenetic model does not consider the effect of oxygen on organic P degradation and organic C/P burial ratios in marine sediments. Organic C/P ratios in sediments deposited under permanently anoxic bottom water conditions can be extremely high (up to 1000) which indicates that bottom water oxygen has a strong effect on P burial. This important observation is not reproduced by the model. Therefore, the authors cannot claim that the Ca effect is stronger than the oxygen effect and that Ca is more important than all other factors regulating P burial (e.g. line 20). They can claim that Ca is important but should not overstate i.e. write that it is more important than anything else.

Response: As stated above in responses to the comments of Reviewers #1 and 2, we have, in the revised version of the manuscript, removed characterization of Ca as the most critical factor regulating P burial. We have instead stressed that Ca levels are a critical boundary condition that must be taken into account in attempts to model the long-term evolution of the P and O cycles.

We have also, in our diagenetic model, parameterized the influence of oxygen on organic C/P ratio for those runs which were coupled to the global C-P-O model (such that anoxic settings can now develop higher organic C/P ratios). We have included these new runs in the revised version of the manuscript.

Moreover, the authors introduce a new approach to calculate porewater pH. This model routine is important because it governs the carbonate ion concentration level in porewater that has a significant effect on CFA formation. The description of this routine is, however, difficult to follow and I am not convinced that the pH calculation fully considers the proton turnover induced by the various reactions that take place in the model (Tab. S2). I do not understand how the proton concentration (H) is treated in the model. H appears as one of the species in Tab. S1. Does this mean that the transport-reaction Eq. 6 is solved for H or is Eq. 14 used to calculate H? If Eq. 14 is applied then the proton turnover induced by the various reactions in Tab. S2 may not be fully considered in the model. The authors should better explain how H is calculated and how the proton turnover is considered in the pH model.

Response: We apologize for the lack of clarity in our description of how pH was calculated. We calculated pH using Eq. 14. And dTP/dt in Eq. 14 was calculated using Eq. 6. $TP = 2CO_2 + HCO_3^- + NH_4^+ + 2H_2PO_4 + HPO_4^{2-} + H_2S + H^+$. So dTP/dt is the summation, of the mass balance function (Eq. 6) of all these species. So, in the full pH equation, all reactions described in Table S2 were fully considered in the dTP/dt term. We have more clearly articulated this in the supplementary information. In other words, we considered all reactions that will affect pH (e.g., organic matter remineralization, inorganic oxidation reactions, reverse weathering and marine weathering, among others). This differs from schemes that consider only organic matter remineralization.

I added a number of additional comments to the supporting information file attached to this review.

Response: We thank the reviewer for this thorough review, and have made the suggested edits to the Supporting Information.

Lines 16-17 How, explain

Response: we have changed that to “The effect of compaction on the solute and solid advection rates was also considered using the method described in ref. 1”.

Why is CO₃ not included in the saturation state calculation but used to calculate the solubility product? Explain

Response: As shown in Equations (4) and (5), CO₃²⁻ is included in the calculation for both the IAP and K_{sp}.

This is strange. CFA should dissolve when porewaters are undersaturated. Please explain how this assumption affects your model results.

Response: we have used this method following the previous models (Dale et al., 2016; van Cappellen and Berner, 1988). CFA is highly insoluble once formed and in general saturation state increases with depth. So for the wide range of conditions explored this simplification seems valid.

This model formulation ignores the effect of dissolved oxygen on sedimentary C/P ratios

Response: We have taken this comment very seriously. We have parameterized the influence of oxygen on the organic C/P ratio in our diagenetic model for the runs coupled with the global C-P-O model. This is the revised version of the manuscript.

What do you mean by differential variable? Above you state that dTP/dt is not calculated in the model and TP is used as an algebraic variable. Please explain

Response: sorry for the mistake. We have changed the relative statement to “Instead of total alkalinity, we used total proton balance (TP) as an equilibrium-invariant and implicit differential variable (whose differentials $\frac{dy}{dt}$ are not treated as a main function and are only used to calculate the $\frac{dy}{dt}$ of the differential variable) to model pH”

I do not understand how the proton concentration is treated in the model. H⁺ appears as one of the species in Tab. S1. Does this mean that you solve the transport-reaction Eq. 6 for H⁺ or do you use Eq. 14 to calculate H⁺?

If you would use Eq. 14 than I would not understand how you consider the proton turnover induced by the various reactions considered in the model (Tab. S2) Please explain how the proton turnover (Tab. S2) is considered in the model

Responses: We have calculated pH using Eq. 14. And dTP/dt in Eq. 14 was calculated using Eq. 6. As $TP = 2CO_2 + HCO_3^- + NH_4^+ + 2H_2PO_4^- + HPO_4^{2-} + H_2S + H^+$, so dTP/dt is the added up of the mass balance function (Eq. 6) of all these species. So, in the full pH equation, all reactions in Table S2 was fully considered in the dTP/dt term. We have added these points to the supplementary information

Line 234, why

Responses: The adsorption of NH₄⁺ was not included in the model as it doesn't influence the P pathways. We have noted this point in the supplementary information.

The measured pH may be wrong due to CO₂ degassing during core retrieval and pH determination. The in-situ pH is probably lower than the measure pH.

Responses: We acknowledge that there may be small errors in pH data, and more precise pH data needed to be measured in the future.

The concentrations of dissolved sulfate, Ca, Mg and Na are much lower than normal seawater values. Why do you apply these low concentrations? Are these low concentrations also applied for the ODP site? Explain and/or use seawater concentrations.

Responses: we have added references for the boundary condition of the FOAM site. The salinity of this site is slightly lower than the normal seawater, so the concentrations of the dissolved components are lower than the normal seawater. For the deep sea site, the values are the same with normal seawater, which are listed in Table S1. Again we have chosen FOAM given that there are just very few sites with all of the relevant data.

Please add another table that defines the rate term that you use for each species considered in the transport-reaction model (Eq. 6 and Eq. 7).

Responses: done. We have added it as Supplementary Table 4.

Reviewer #3 (Remarks to the Author):

Review of Zhao et al., for Nat. Comm.

Zhao et al. explore an interesting idea in this contribution that changes in seawater Ca²⁺ concentrations over geologic time could have affected the regeneration of P from sediments. Their idea is straightforward and sound — Phosphorous is trapped in sediments either as organic matter, bound to iron, or precipitated as apatite (carbonate fluorapatite [CFA] dominantly). Oversaturation for CFA occurs in sediments when the rate of phosphorus release from organic carbon degradation outpaces the diffusional loss of P to the subsurface allowing CFA to become oversaturated and precipitate. Seawater Ca²⁺ concentrations have varied by a factor (we think based on fluid inclusion data) of about 2-3 over the Phanerozoic. Thus, it is reasonable to explore the idea that elevated Ca²⁺ levels in past oceans could have served to make it easier to precipitate CFA in the sediments for a given phosphorus level in the oceans. In terms of the most basic idea of the paper, it is sound and interesting and worthy of exploration and will have useful impact on experts in the field.

However, how the authors go about exploring this idea in a quantitative sense I could not follow nor evaluate in the main text. Three separate models are developed. Two are sedimentary reaction-diffusion equations and these are then connected to a global carbon cycle model to explore the results over the past 500 million years. This involves 96 separate differential equations (many of which are coupled) and requires assumptions about rate constants and the like — note I just counted up the number of equations in the SI, some may be redundant, but regardless, the number is high. However, none of the equations used are presented in the main

text nor are they discussed. None of the assumptions made are discussed. How various rate constants were chosen is not discussed in the main text. Rather all of this is placed in the supplementary information. As a result, in this contribution, the supplementary information, rather than adding extra information to provide a curious reader additional context for the main manuscript, is required reading and must be read along with the main manuscript both to understand exactly what it is the authors have done, and to be able to evaluate the results. In other words, in my reading, the main manuscript does not stand on its own. Based on this, I cannot recommend the paper as written to Nature Communications as a reader. This is not a criticism of the paper's substance (which I like, though I found many aspects confusing due to the large amount of info in the SI). Instead, I would encourage the authors to submit to a journal like AJS or GCA in which their models are fully explained and contextualized, assumptions made clear, and model space explored. In other words, the manuscript, as it stands, does not stand on its own and requires careful co-reading of the SI to follow and evaluate the model outputs.

Response: We acknowledge that our Supplementary Information is rich in information concerning our modeling approach. We have moved much of the text in Supplementary Information to the Method part of the main text. We consider that we have robustly and convincingly demonstrated our central point—that marine Ca concentrations will play a key role in marine P burial—with a very simple model as well as with our ‘extended’ (i.e., more realistic) diagenetic model. Our very simple model can be easily understood by anyone familiar with diagenetic models (and could be easily coded in a few hours from the equations we present and the parameters given in the Tables). The basis for both of these models (the general diagenetic equation; cf. Berner, 1971) is very well established, as is the core of our model. In our opinion the three most intuitive ways to understand a model are 1) to see the core equations, 2) to see the constants and boundary conditions in a table and 3) to be able to look at the actual code. All of these options are possible for our manuscript—the equations and tables are clearly laid out and described in the Supplementary Information and we will make the code publicly available upon publication.

All authors of this paper have tremendous respect for disciplinary journals, including the *American Journal of Science* (the senior author is, in fact, an AE at *AJS*). But we also are willing to admit the shortcomings of *AJS* in terms of ability to reach a broad audience—unfortunately, far fewer people will read this paper if it is published at *AJS* instead of *Nature Communications*. And we think that our central conclusions represent a very exciting advance in the community's understanding of the secular co-evolution of biogeochemical cycling and the biosphere, and one ideally suited to the broad readership provided by *Nature Communications*. Moreover, we feel that both the structure and mandate of *Nature Communications* papers—a succinct presentation of a topical and broadly interested question and robustly supported conclusions, coupled to Supplementary Information offering a more detailed explication of methods used—is actually ideally suited to presentation of our results, and to offering proper empirical and methodological support for our conclusions. Therefore, we feel that *Nature Communications* is an appropriate journal for this paper—and, in fact, ideally suited to presentation of our central findings.

Specific comments:

(1) *Something that was not clear to me in reading the paper is how the calculations were done.*

Specifically, is the dissolved phosphorus content of the ocean held constant as Ca^{2+} is varied for example? Does raising Ca lower this (given that there is less P removed from the sediments I assume it must). Given that the system, I assume, reaches a steady state in which P inputs equal P outputs, does the regeneration really matter if what goes in to the ocean goes out on geologic timescales? I.e., when a perturbation is forced (a change in Ca^{2+}) how long does the system take to relax such that inputs equal outputs. Figure 4 appears to indicate, at least to me, an immediate response. Figure 3 shows some of this variation, but I would like to see the relationship between Ca vs PO_4 in seawater for steady state solutions.

Response: We have reworded several portions of the text in order to more clearly explain how these processes are treated in our models. In our coupled C-P-O cycle model, P concentration (e.g., marine reactive phosphorus) varies. The P weathering flux is, at steady state, equal to the P burial flux at steady state. The regeneration rate therefore is significant because it can be treated as the speed at which P cycles through the marine biosphere. Thus high regeneration rates will mediate increases in the size of the marine biosphere.

As requested by the reviewer, we have created and included a new figure to more clearly demonstrate the influence of Ca on seawater $[\text{PO}_4]$ and atmospheric oxygen levels (Fig. 5). These results indicate that an increase in seawater dissolved Ca concentration will, through increased precipitation of CFA, lead to decreased marine dissolved P levels. The response of the P cycle to changes in dissolved Ca concentration is rapid, relative to the timescales required for a change in seawater Ca. An increase in seawater dissolved Ca will induce an increase in CFA burial but a decrease in organic P burial. The total P burial flux does not change, however, as it is equivalent to the P weathering flux, which we fixed in this run.

(2) In figure 4, we see PCFA/Preactive plotted. How does P reactive change with time vs. P total? And how do C/P ratios change. I found this comparison confusing as without knowing how the denominator (P reactive) is changing with time, it is unclear to me if there is actually less P available back released to the ocean, or not. Something else that concerned me about this plot is that the PCFA/Preactive has the highest uncertainties for the modern samples, and the lowest uncertainties for the ancient samples. Is there an aliasing going on here that we should worry about?

Response: The ratios of Reactive P to Total P in modern sediments are shown in Fig. S8. These are nearly constant, with a slight decrease during the last 10 myr. As Total P is the sum of reactive P and detrital P, this ratio may provide some information regarding extents of terrestrial P weathering. However, this does not straightforwardly provide information about P diagenetic pathways in sediments. Variations in the non-normalized reactive P burial flux are unfortunately challenging to deduce, given large uncertainties regarding sedimentation rates. Aliasing is unlikely possible because the amount of samples for modern samples are large (>30 for each time bin), and the distribution of data is nearly even across time.

(3) Figure 4b is important as it shows PCFA burial changing with time that inversely correlates with $p\text{O}_2$ (indicating a coupling). But how does total P burial change with time in the model? I'm assuming this is constant (or at least that P weathering equals P burial on ~million year

timescales). Does the P flux to oceans change? Is the C/P ratio of sediments changing? The graph implies that PCFA is important. But does it really matter what PCFA fluxes are vs. total fluxes? Alternatively, I would think the P regeneration rate is really what matters (i.e., whether you get P out of sediments, but leave Corg behind). This will impact C/P ratios.

Response: The total P burial and total P weathering fluxes are shown in Fig. S4. Total P burial is controlled by the P weathering flux on geologic time scales as the residence time of P (~70000yr) is relatively short. In our default run of the coupled C-P-O model, the total P burial changes with time. Input fluxes change foremost with land plant-assisted weathering. In our model (following COPSE and GEOCARB) the rise of land plants plays a stronger role in controlling P weathering fluxes than do orogenic events). Enhanced CFA burial will lead to more P retention, lower marine dissolved P concentrations, and less organic carbon burial (Fig. 5 and Supplementary Fig. 5). This surely will impact the C/Reactive P ratio, which is shown in Supplementary Fig. 5.

Yours sincerely (on behalf of all the authors),

Mingyu Zhao

Reviewers' comments:

Reviewer #1 (Remarks to the Author):

I made three major criticisms of this paper. The first and third were linked, and were of the authors' possibly overstated conclusions that Ca concentration was the dominant factor controlling marine P burial over geological time. The authors now agree in this revision that Ca concentration is not the most important control. So this point is now resolved, but it is resolved by removing the most impactful finding of the paper. I question now whether these results are interesting enough for the general readership of Nature Communications, but I will leave that decision for the editor.

If the editor still wishes to publish this paper, or when they submit elsewhere, then my second point should also be addressed properly. The point I made was that their Phanerozoic global scale model does not consider many of the important processes in O₂ regulation, and thus was no use for making conclusions on what level of O₂ variation could be achieved by changing Ca concentration. The model they use was designed to explore feedbacks in isolation, not to make Phanerozoic pO₂ predictions. I suggested instead using a comparison where the relative changes in O₂ caused by changing Ca, bioturbation etc. could be assessed, and I am glad that they have done this in figure 5. But there are problems with this:

1. There is almost no mention of figure 5 in the text. I am left with many questions about this figure: Why the different time scales for each experiment? Why is one effect so much stronger when they look to be a similar magnitude in figs 2 and 3? Why the odd dynamics for the solid line? what is driving this reversal? This is all very important and not explained.

2. I recommended that the Phanerozoic figure be removed, and it has not been. The authors rebuttal notes that the qualitative relationships are robust even if the model is too simple. This is true, but my point was about the quantitative results, and the lack of data used to force the model, and neither have been addressed. I regret that my previous point was not given in enough detail so here are specific lines:

142: "We further forced this global carbon cycle model with previously estimated, independently derived values for seawater DIC, seawater dissolved Ca and Mg concentrations, seawater pH, and bioturbation intensities through the Phanerozoic".

- None of this information is plotted or described, so the model is still very opaque.

152: coupled model shows that "Phanerozoic CFA burial fluxes largely follow contemporaneous variations in seawater dissolved Ca concentrations"

- But this is because many other important processes have been left out. What about erosion, sedimentation and organic carbon burial for example? What about the weathering flux of P? How does anoxia change in the model?

156: "Our coupled global model also suggests that major swings in CFA and P burial result in similarly sizeable swings in atmospheric oxygen levels."

-The swings are on different orders of magnitude, not the same size.

158: "Variations in marine Ca concentrations could have caused shifts in atmospheric pO₂ from less than 10% to greater than 25% (v/v), encompassing essentially the full range of values previously predicted for the Phanerozoic as a whole".

-But if the model contained other important processes (fire feedback, sulfur cycle etc.) these numbers would be much less.

163: "Fluid-inclusion data suggest an increase in seawater dissolved Ca concentrations during the early Cambrian 11". -The inclusion data does not show this. There is only one data point pre-

Devonian.

178: "Our results do not preclude the possible importance of the rise of land plants in shaping atmospheric pO₂. However, they suggest that seawater dissolved Ca concentration may have been an equally important factor influencing surface oxygen levels over the duration of the Phanerozoic"
-As earlier, Ca is very important in this model because not much else is considered. The rise of plants is not considered in this model in any detail so the relative importance cannot be assessed.

193: "Within this traditional framework, atmospheric pCO₂ and pO₂ should be positively correlated over major tectonic cycles during Earth's history"
- I do not recognise this 'traditional framework'. This statement is only true if spreading rates change while everything else remains fixed, which is not a good representation of the way the system works. Biological weathering enhancements should increase O₂ and reduce CO₂?

194: "This theoretical paradigm, however, clashes with the broadly negative correlation observed between pCO₂ and pO₂ in reconstructions of Phanerozoic atmospheric composition"
- The latest estimates I can find for Phanerozoic O₂, which include some of these authors and are in this journal, show a decrease from 200 Ma to present, which does correlate with CO₂ decrease.

Reviewer #2 (Remarks to the Author):

The authors carefully considered all of the reviewers' comments in the revision of the manuscript. It is now in very good shape and can be published as submitted.

Klaus Wallmann

Reviewer #3 (Remarks to the Author):

Enclosed is my second review of Zhao et al.

Zhao et al. have addressed some of my comments. I still find the paper unclear in many parts with too much of what was actually done present in the SI or methods. However, I appreciate the authors moving more of the paper to the methods sections as this makes much of the paper more transparent.

As I stated in my original review, the paper is interesting and the overall basic message is clear. However, I still find myself having trouble understanding what precisely the authors did and more importantly their assumption. This is because a large number of models and assumptions are being made that, when reading the main text, are not clear. I appreciate the author's response to this — that I can just explore the issues with the model myself (as stated in the response). However, as far as I could tell, the code would not be published alongside the paper, but rather would only be available upon request and so I cannot at this time do this.

In summary, I personally think this paper would do better and have more impact in a longer-format journal. The authors disagree, and I think this is a fair disagreement. I appreciate the authors comments that more people will read the paper if published in Nat. Com and thus that their central message will get across to a larger body of people. The authors have tried to clarify things and have responded in earnest to the reviews. As such, I do not believe I need to see it again as my main reservation is not one of whether or not what they have done is correct or interesting, but rather the great reliance on the SI . If the editors think it is sufficiently broad and accessible, then that is good

enough for me.

Main comments:

(1) This paper is a modeling paper with multiple models presented. However, the code used for the calculation is not provided such that a reader cannot inspect what they have done or attempt similar sensitivity tests. It is stated that the code can be requested from the authors. However, I do not know the state of the code and if, at this moment, someone not involved in the work could actually use it — i.e. is there clear documentation associated with the code? At this point, I believe the authors should include all code used as part of this study for all models discussed (including that coupled to the global carbon cycle model) with appropriate documentation. I believe these should be provided as part of the SI.

(2) There remains a deeply unclear aspect of the work. The authors, in their response, acknowledge that total phosphorus input equals total phosphorus output. Their models, through changes in Ca^{2+} affect the dissolved phosphorus content of the oceans, which they then take to limit productivity (and thus corg burial). As total phosphorus burial remains constant for any geologically relevant time frame, the parameters that matter for our understanding of how Ca^{2+} affects biogeochemical cycling is the overall C/P ratio of sediments following diagenesis. Put another way, what we need to know is the percent of P recycled from sediments back to the water column vs. the amount of organic carbon deposited and respired. And we need to know how these change together as a function of Ca^{2+} concentrations. This is presumably built into the model (C/P ratios), but such numbers would be useful to present for their given scenarios as this is a commonly measured parameter (C/P ratio). Regardless, there are numerous statements about important changes to P burial by Ca^{2+} . This is a only sort of correct as far as I understand it as net phosphorus burial is a constant (unless weathering changes). The authors are well aware of all of this, but it would be helpful if they were careful in their language on the matter and provided C/P ratios to help guide the reader and provide testable numbers. I'll note in my last review, I recommend the authors include C/P ratios, but they did not address this.

Specific comments:

Line 48 — Stoichiometry: Does the precise stoichiometry matter? If more or less Co_3 is put in does it have an effect? I do not see a reference for this chosen stoichiometry.

Line 51 — I would substitute 'should' for 'could'. I'll note this sort of statement is common and I advise the authors to think carefully about what they have inferred vs. demonstrated.

Line 60 — What is an 'intrinsic rate law'. Perhaps I am ignorant, but I have never heard of such a thing, nor do I see a definition. I did a quick google scholar search for this term, and came up with 22 references total when searched in quotes. Regardless, I don't know what the authors are talking about. It seems like rate law is just proportional to the degree of undersaturation vs. supersaturation, which is a common model form. Are you sure the rate law is order 1 vs. some other order as is common for calcite dissolution? How well does your model fit the experimental data?

Line 66 — What are the key simplifying assumptions you have made for you basic model exercise. Is there anything the author should worry about here that was a big assumption you made? Does the rate of organic carbon oxidation matter for example? I.e., if you speed up the rates of remineralization so more happens at the surface, do you not make CFA at all?

Lines 73-77 — what are calibrating? This implies you do not know certain terms and are doing a fitting exercise to find them. Is FOAM really the best place to do it? If you take your parameters and then use them in another site where things are well understood, do they give appropriate answers?

Line 114 — does bottom water temperature play a role? Are you assuming a temperature dependence both for the saturation state as well as the kinetics of dissolution and reprecipitation? This is important given the large changes in bottom water temperatures over the Cenozoic.

Line 121 : Is the data controlled for depth, bottom water temperature, and all other variables. Are all sites from the different ocean basins present for all time? This is an important dataset for the paper, but it is largely unexplored and explained. If you look at the Pacific vs. Atlantic, do they have the same behavior over time? Why does this only go back to 80 million years?

Lines 121: If you are going to state something 'correlates strikingly', you should provide the R2 of the correlation and actually show the cross plot.

Line 129: I don't think this is a 'test' as opposed to exploring with the model

Line 140: Is it DIP that really matters for GPP? Or rather is it the shallow/deep water mixing flux. DIP is very low in the surface ocean where it is limiting. But there can still be high GPP if the flux from below is high enough driving by extensive recycling. This raises an important point — you appear to be modeling marine primary productivity levels including both GPP and NPP. Please provide those values so we can see if they are reasonable. Also, what happens to deep-ocean O2 concentrations as you lower GPP and NPP at low DIP? Will we lose our anoxic zones? Or do you just have pO2 change to compensate — i.e., you are keeping dissolved ocean O2 the same.

Complex model — provide the estimated C/P ratios for the model run and compare to data? This would be quite helpful.

Figures

Figure 1: Do we have an external data on what a reasonable P regeneration percent should be? If we look at the modern calculated values, they are all predicted in the model to be about 50%. So I should expect, based on this, that typical C/P ratios of sediments (assuming they have chosen typical values) should be around 200-250. This seems correct to me. What about for the anoxic endmember? Is this consistent with data from those basins? If you look at C/P of 100 million year old sediments, are they different given you large predicted change in the regeneration rate?

Figure 2: Is burial efficiency here just one minus the regeneration rate from figure 1? If so, perhaps use the same units. I would like to see the predicted C/P ratios for these curves as these can be more easily related to actual data to check whether what we are seeing is reasonable. Such a ratio is inherently calculated by the model.

Figure 3: Again I would provide all data in terms of a C/P ratio where possible. Additionally, something I find confusing is that total burial of P is a constant (it is just the weathering flux). So when the amount of CFA in sediments grow, what happens to the other sinks? I.e., where does the additional P come from or go to in the sediments as CFA concentrations change.

Figure 4: The model is built on the FOAM site, which is not a deep sea sediment. Is this an appropriate comparison? Why not use deep-sea sediments for the parameterization. Please break down the PCFA/Preactive based on the cores they came from and basins. It would be nice to make sure there is not an aliasing in the record. Finally, please show these a C/P ratios. I would like to make sure those change as would be predicted.

Figure 5 Is the pO2 actual percent (i.e. taking into account 0.8 bars of N2) or is it a pressure.

Figure 6: Again, it would be useful to see a C/P ratio as this is something that is measurable and presumably predicted by the model given absolute corg burial fluxes are being used to determine pO2.

NCOMMS-19-07813A: Responses to Reviewer Comments

Reviewer #1:

I made three major criticisms of this paper. The first and third were linked, and were of the authors' possibly overstated conclusions that Ca concentration was the dominant factor controlling marine P burial over geological time. The authors now agree in this revision that Ca concentration is not the most important control. So this point is now resolved, but it is resolved by removing the most impactful finding of the paper. I question now whether these results are interesting enough for the general readership of Nature Communications, but I will leave that decision for the editor.

We feel strongly that this is not a fair reading of the paper. And, with all due respect, that is not a useful way of thinking about global biogeochemical processes. Through all of the revisions we have embraced the idea that the global P cycle is going to be controlled by a wide range of factors. Our analysis indicates that Ca will exert as significant or a more significant impact on the global P cycle as other commonly invoked drivers of global biogeochemical change (e.g., land plants, bioturbation). This is a critical observation, as the Ca cycle is typically not considered in evaluations of what factors have changed the scope of the biosphere through time. We feel strongly that this simple message—that Ca is a significant driver of the coupled P-C-O cycle—is novel, important and will be impactful.

We never stated, as the reviewer suggests, that “Ca concentration is not the most important control” on the global P cycle and we don't think that this sentiment reflects either our analytic results or our discussion of stratigraphic and model data. We also urge that absolutist is neither helpful nor reflective of the contents of our study. Although we are extremely grateful for the reviewer's time, we found this review to be unnecessarily antagonistic.

If the editor still wishes to publish this paper, or when they submit elsewhere, then my second point should also be addressed properly. The point I made was that their Phanerozoic global scale model does not consider many of the important processes in O₂ regulation, and thus was no use for making conclusions on what level of O₂ variation could be achieved by changing Ca concentration. The model they use was designed to explore feedbacks in isolation, not to make Phanerozoic pO₂ predictions.

Thank you very much for this suggestion. We have moved that part to the supplementary information.

I suggested instead using a comparison where the relative changes in O₂ caused by changing Ca, bioturbation etc. could be assessed, and I am glad that they have done this in figure 5. But there are problems with this. There is almost no mention of figure 5 in the text. I am left with many questions about this figure: Why the different time scales for each experiment? Why is one effect so much stronger when they look to be a similar magnitude in figs 2 and 3? Why the odd dynamics for the solid line? what is driving this reversal? This is all very important and not explained.

Sorry for the problems. We have added more discussion of Figure 5 to the main text. We have also, in the revised version of the manuscript, used the same time scale for each model simulation.

In Figure 2 and 3, we did not assume that the organic C/P ratio varied as a function of seawater oxygen. But in Figure 5, we applied that relationship (and this was noted). This results in Ca driving

a more pronounced change, relative to bioturbation, in the global P cycle. We have now added a supplementary figure to make this clearer (Fig. S4).

The reversal in oceanic reactive P reservoir is due to the delayed decrease in atmospheric oxygen and decrease in iron-bound P burial. The delayed decrease in atmospheric oxygen is simply a reflection of its long residence time. We have also clarified this point in the main text.

2. I recommended that the Phanerozoic figure be removed, and it has not been. The authors rebuttal notes that the qualitative relationships are robust even if the model is too simple. This is true, but my point was about the quantitative results, and the lack of data used to force the model, and neither have been addressed. I regret that my previous point was not given in enough detail.

We have moved this figure to the supplementary information—where we can provide more contextual information.

However, we would like to stress that we have not stated that the model is too simple—we feel strongly that this iteration of the model is sufficiently complex to illustrate our central point—that the Ca cycle exerts a strong influence on atmospheric oxygen concentrations. We purposefully did not use the most up-to-date highly parameterized version of COPSE (or an equivalent model) to try to refine forward-model based Phanerozoic pO_2 estimates. We used a simpler model in order to more straightforwardly explore the strength of the Ca feedback relative to other key, previously recognized feedbacks on the C-O cycle.

142: “We further forced this global carbon cycle model with previously estimated, independently derived values for seawater DIC, seawater dissolved Ca and Mg concentrations, seawater pH, and bioturbation intensities through the Phanerozoic”.

- None of this information is plotted or described, so the model is still very opaque.

The values were listed in Supplementary Table 7 and there is a reference to that table in this section of the text. However, we note this aspect of the manuscript is now confined to the supplement.

152: coupled model shows that “Phanerozoic CFA burial fluxes largely follow contemporaneous variations in seawater dissolved Ca concentrations”

- But this is because many other important processes have been left out. What about erosion, sedimentation and organic carbon burial for example? What about the weathering flux of P? How does anoxia change in the model?

We agree that we do not consider all processes affecting the global P cycle, nor have we claimed that we do so. We have made it very clear that we are not trying to explore the effects that major orogenic episodes, for instance, may have had on the global P and C cycles (and thus the effects of erosion and sedimentation).

Organic carbon burial is coupled between the global C-P-O model and the diagenetic model.

The weathering flux of P was parameterized on the assumption of plant-assisted weathering (following COPSE). The change of anoxia was calculated using equation (C33), shown in Supplementary Table 10.

All of this discussion has now been moved to the SI.

156: *“Our coupled global model also suggests that major swings in CFA and P burial result in similarly sizeable swings in atmospheric oxygen levels.”*

-The swings are on different orders of magnitude, not the same size.

We have changed “similarly sizeable” to “major”. This portion of the manuscript is now in the Supplementary Information.

158: *“Variations in marine Ca concentrations could have caused shifts in atmospheric pO₂ from less than 10% to greater than 25% (v/v), encompassing essentially the full range of values previously predicted for the Phanerozoic as a whole”.*

-But if the model contained other important processes (fire feedback, sulfur cycle etc.) these numbers would be much less.

This section has been moved to the supplement. But to make our approach clearer, we have changed this sentence to:

“Considering a global oxygen cycle with only redox and nutrient feedbacks, variations in marine Ca concentrations could have caused shifts in atmospheric pO₂ from less than 10% to greater than 25% (v/v).”

We also added:

“There are, of course, a wide range of factors which may influence O₂ levels (e.g., paleoceanographic circulation, sulfur cycling, fire-associated feedbacks and orogenic uplift) not all of which are considered here.”

163: *“Fluid-inclusion data suggest an increase in seawater dissolved Ca concentrations during the early Cambrian 11”. -The inclusion data does not show this. There is only one data point pre-Devonian.*

We have reworded this sentence to “Fluid-inclusion data and carbonate mineralogical records are commonly interpreted to suggest an increase in seawater dissolved Ca concentrations during the early Cambrian” and we have added more references in support of this statement.

178: *“Our results do not preclude the possible importance of the rise of land plants in shaping atmospheric pO₂. However, they suggest that seawater dissolved Ca concentration may have been an equally important factor influencing surface oxygen levels over the duration of the Phanerozoic”*

-As earlier, Ca is very important in this model because not much else is considered. The rise of plants is not considered in this model in any detail so the relative importance cannot be assessed.

We included (in this and previous versions of the manuscript) plant-assisted weathering, using a previously proposed feedback (see equation S15). We have revised this sentence for clarity, and this section has been moved to the SI. However, our point was not that other factors are not important drivers of shifts in the global P cycle—our point is simply that forcing from Ca is comparable in strength; something that has not been previously recognized.

193: *“Within this traditional framework, atmospheric pCO₂ and pO₂ should be positively correlated over major tectonic cycles during Earth’s history”*

- I do not recognise this ‘traditional framework’. This statement is only true if spreading rates

change while everything else remains fixed, which is not a good representation of the way the system works. Biological weathering enhancements should increase O₂ and reduce CO₂?

We have removed this section.

194: “This theoretical paradigm, however, clashes with the broadly negative correlation observed between pCO₂ and pO₂ in reconstructions of Phanerozoic atmospheric composition”

- The latest estimates I can find for Phanerozoic O₂, which include some of these authors and are in this journal, show a decrease from 200 Ma to present, which does correlate with CO₂ decrease.

We have removed this section. And it's a fair point that, although previous model outputs (e.g., Berner's last GEOCARB publications) do show a broadly negative correlation, pO₂ and pCO₂ trajectories are less obviously decoupled in the recent publication by Krause et al. (2018).

Reviewer #2:

The authors carefully considered all of the reviewers' comments in the revision of the manuscript. It is now in very good shape and can be published as submitted.

Klaus Wallmann

Thank you for this positive review, and thoughtful previous reviews of this manuscript.

Reviewer #3:

Enclosed is my second review of Zhao et al.

Zhao et al. have addressed some of my comments. I still find the paper unclear in many parts with too much of what was actually done present in the SI or methods. However, I appreciate the authors moving more of the paper to the methods sections as this makes much of the paper more transparent.

As I stated in my original review, the paper is interesting and the overall basic message is clear. However, I still find myself having trouble understanding what precisely the authors did and more importantly their assumptions. This is because a large number of models and assumptions are being made that, when reading the main text, are not clear. I appreciate the author's response to this — that I can just explore the issues with the model myself (as stated in the response). However, as far as I could tell, the code would not be published alongside the paper, but rather would only be available upon request and so I cannot at this time do this.

In summary, I personally think this paper would do better and have more impact in a longer-format journal. The authors disagree, and I think this is a fair disagreement. I appreciate the authors comments that more people will read the paper if published in Nat. Com and thus that their central message will get across to a larger body of people. The authors have tried to clarify things and have responded in earnest to the reviews. As such, I do not believe I need to see it again as my main reservation is not one of whether or not what they have done is correct or interesting, but rather the great reliance on the SI. If the editors think it is sufficiently broad and accessible, then that is good enough for me.

We thank the reviewer for this thoughtful review, and their support for the publication of this manuscript. In order to address the point raised by the reviewer—the transparency and accessibility of the paper—we have removed some aspects of the paper to make it more readily ‘digestible’ (e.g., the discussion of $p\text{CO}_2$ and $p\text{O}_2$ correlations or lack thereof; see above response to Reviewer #1).

We are also still confident that our central message—that the Ca cycle will exert a strong role on global P cycle—is simple and suitable for a broad audience. We demonstrate this point with a very simple diagenetic model as well as with a more complex diagenetic model. We do, however, disagree that it is an inherent flaw for a paper to contain detailed information about a model and its mathematical underpinnings in supplementary form. In our view, to provide very detailed information concerning the ‘nuts and bolts’ of a model (that are not necessary to understand or support our central findings but which may be of interest to and, in future, utilized by readers wishing to employ a similar approach) is, in fact, an ideal use of a paper’s Supplementary Information.

We would be very happy if, if we put this paper in a long-format modeling-focused journal (like *AJS*), as many people would read this paper as would if it were published in *Nature Communications*. But of course this is not the case. We feel strongly that our message is important and broadly applicable to researchers from many different disciplines (e.g., biogeochemical cycling, Earth-system evolution, paleontology, geobiology, Precambrian geology, oceanography, climate sciences) concerned with the primary controls upon carbon and nutrient cycling and how these have evolved through time, in a manner that transcends the boundaries of any single discipline. Moreover, the results we present bear directly upon interpretations communicated by previous studies published in similarly high-impact and multidisciplinary journals; we consider that publication of our study in a journal that will reach a similarly broad readership is essential in order to fully engage in this debate (on the primary controls on carbon cycling and oxygenation) and to make all aspects of this debate broadly accessible. We therefore think that the broad readership and exposure provided by *Nature Communications* are ideally suited to communication of these findings and this message.

Main comments:

(1) This paper is a modeling paper with multiple models presented. However, the code used for the calculation is not provided such that a reader cannot inspect what they have done or attempt similar sensitivity tests. It is stated that the code can be requested from the authors. However, I do not know the state of the code and if, at this moment, someone not involved in the work could actually use it — i.e. is there clear documentation associated with the code? At this point, I believe the authors should include all code used as part of this study for all models discussed (including that coupled to the global carbon cycle model) with appropriate documentation. I believe these should be provided as part of the SI.

We have included a lengthy SI because we wish to be transparent about the core equations and data utilized in the model. We have listed all of the parameters used in the model and we have stressed that our diagenetic model was built using the ReacTran package. So it would be fairly simple, from this information, for someone to recreate our diagenetic model. Our intention was to post the code upon acceptance of this manuscript. But, in light of the fair point the reviewer makes, we have included the code in this submission. And, as now noted in the revised manuscript, we will upload our model to GitHub (with proper documentation) upon publication.

(2) There remains a deeply unclear aspect of the work. The authors, in their response, acknowledge

that total phosphorus input equals total phosphorus output. Their models, through changes in Ca²⁺ affect the dissolved phosphorus content of the oceans, which they then take to limit productivity (and thus corg burial). As total phosphorus burial remains constant for any geologically relevant time frame, the parameters that matter for our understanding of how Ca²⁺ affects biogeochemical cycling is the overall C/P ratio of sediments following diagenesis. Put another way, what we need to know is the percent of P recycled from sediments back to the water column vs. the amount of organic carbon deposited and respired. And we need to know how these change together as a function of Ca²⁺ concentrations. This is presumably built into the model (C/P ratios), but such numbers would be useful to present for their given scenarios as this is a commonly measured parameter (C/P ratio). Regardless, there are numerous statements about important changes to P burial by Ca²⁺. This is a only sort of correct as far as I understand it as net phosphorus burial is a constant (unless weathering changes). The authors are well aware of all of this, but it would be helpful if they were careful in their language on the matter and provided C/P ratios to help guide the reader and provide testable numbers. I'll note in my last review, I recommend the authors include C/P ratios, but they did not address this.

We agree that the percent of P recycled from sediments—at a given TOC flux—dictates the effect of varying Ca concentrations. What we have tried to make clearer is that, at a given TOC flux, P is preferentially retained within the sediments when boundary-condition (i.e., bottom-water) Ca concentrations are higher. Our basic argument is that more CFA burial at high seawater Ca concentrations will induce less organic P burial and less P recycling from sediments back to the water column (shown in our manuscript and model as high burial efficiency). This will, in turn, result in less organic carbon burial and a drop in atmospheric oxygen.

That being said, we agree that showing the $C_{\text{org}}/P_{\text{reactive}}$ values would be helpful to readers and provide an additional framework for comparison with the findings of other studies that use this nomenclature. We have therefore added these ratios to Fig. 2, Fig. 3, Fig. S4 and Fig. S6.

Specific comments:

Line 48 — Stoichiometry: Does the precise stoichiometry matter? If more or less Co₃ is put in does it have an effect? I do not see a reference for this chosen stoichiometry.

We have included a supporting reference on this point. We used an average stoichiometry for carbonate fluorapatite, based on what has been previously documented from marine sediments. More or less carbonate ion will not have a strong influence on the Ca effect, as the Ca stoichiometry is always close to 10.

Line 51 — I would substitute 'should' for 'could'. I'll note this sort of statement is common and I advise the authors to think carefully about what they have inferred vs. demonstrated.

We have accordingly revised this statement.

Line 60 — What is an 'intrinsic rate law'. Perhaps I am ignorant, but I have never heard of such a thing, nor do I see a definition. I did a quick google scholar search for this term, and came up with 22 references total when searched in quotes. Regardless, I don't know what the authors are talking about. It seems like rate law is just proportional to the degree of undersaturation vs. supersaturation, which is a common model form. Are you sure the rate law is order 1 vs. some other order as is common for calcite dissolution? How well does your model fit the experimental data?

To avoid confusion, we have deleted “intrinsic.” Given that experimental data often cannot, alone, be directly applied to conditions in natural systems, we calibrated the reaction rate constant (through a scaling with the CFA saturation state) using data from modern sediments. The order in rate law for CFA formation may vary under different conditions (just as calcite does). But, as a start, we use a 1st order rate law (following what is used in most attempts to model calcite formation). A higher order will, increasingly, amplify the effect of Ca. We have clarified this point in the supplementary information.

Line 66 — What are the key simplifying assumptions you have made for your basic model exercise. Is there anything the author should worry about here that was a big assumption you made? Does the rate of organic carbon oxidation matter for example? I.e., if you speed up the rates of remineralization so more happens at the surface, do you not make CFA at all?

Our simplified model includes only two reactions: the decomposition of organic matter and the formation of CFA. The organic carbon oxidation rate is from a shallow-water setting (our baseline shallow-water model; which matches the FOAM site). *The goal of this model is only to straightforwardly demonstrate that $[Ca^{2+}]$ will strongly affect CFA formation.* The goal of the more complex diagenetic model is to illustrate that, even when considering a full suite of reactions—a model with all factors that could influence P burial within the sediment pile— $[Ca^{2+}]$ still exerts a very strong control on P burial.

In the simple model, when we increase the remineralization rate by 10 times, we still observe CFA formation. The extent of the shift in CFA formation with the rate increase depends, of course, on TOC flux, DIC and phosphate concentration of bottom water. We now show this effect as the new Supplementary Figure 1.

Lines 73-77 — what are calibrating? This implies you do not know certain terms and are doing a fitting exercise to find them. Is FOAM really the best place to do it? If you take your parameters and then use them in another site where things are well understood, do they give appropriate answers?

We have calibrated the reaction rate constant for CFA formation. We have clarified this point in SI. The FOAM site is, as numerous previous work (some of which we cite in the manuscript) has shown, a reasonable proxy for a typical shallow-water setting. More importantly, we selected FOAM to evaluate the robustness of our model given that it is a site from which an usually complex dataset has been previously collected and published, including empirical data on bioturbation intensities and sediment mixed layer depths; various porewater chemical species and complete P speciation such as iron-bound P, authigenic P and organic P; a complete redox balance has been determined, and various diagenetic processes have been extensively investigated (see Figure S2).

Somewhat to our surprise, we have not been able to find a better site with detailed geochemical data and information on bioturbation, which is necessary for the development and testing of our model. We have clarified these points in the SI.

We have also applied our model to a ‘typical’ deep-sea site, ODP site 1226. Various porewater chemical species and complete P speciation have been previously measured for this site, making it amenable to our model-based assessment (see Fig. S3). Importantly, our model was also able to reproduce these deep-sea profile data. This information is also included in the SI.

Line 114 — does bottom water temperature play a role? Are you assuming a temperature dependence both for the saturation state as well as the kinetics of dissolution and reprecipitation? This is important given the large changes in bottom water temperatures over the Cenozoic.

This is a great question. Given the complexity already present in this model, we have not included a temperature dependence for all of the reaction terms. However, we have a version of the model that has fully parameterized the effect of temperature on the reaction rates and K_{sp} , the activity coefficient. The effect of temperature on P burial is muted relative to the effect of redox state, TOC fluxes, and $[Ca^{2+}]$. Therefore, and in light of the extra complexity incorporating temperature effects would entail (and doing so would, we surmise, be counter to the requests of this and other reviewers who have requested greater simplicity in the model and its explication), we prefer to retain to not incorporate temperature dependence into this study, and to instead develop this more fully in a separate paper (and one that explores a range of factors besides P burial).

Line 121: Is the data controlled for depth, bottom water temperature, and all other variables. Are all sites from the different ocean basins present for all time? This is an important dataset for the paper, but it is largely unexplored and explained. If you look at the Pacific vs. Atlantic, do they have the same behavior over time? Why does this only go back to 80 million years?

We have added further contextual environmental information, including depth and bottom water temperature, to Table S7. Note that bottom-water temperature data are unfortunately lacking from some sites. We have now plotted Pacific and Atlantic data separately in Figure 4. These two profiles are characterized by the same trend. Continuous P speciation records only exist from IODP cores (i.e., the last 80 myr). The empirical data for seawater Ca concentrations prior to 80 myr are also rare. We hope that future work will continue to expand the temporal range of these records.

Lines 121: If you are going to state something ‘correlates strikingly’, you should provide the R2 of the correlation and actually show the cross plot.

We have changed “correlates strikingly” to “corresponds well with.” Also the model outputs can explain the observed shift—and it is that correspondence that we consider to be most important.

Line 129: I don’t think this is a ‘test’ as opposed to exploring with the model

We have changed “test” to “explore.”

Line 140: Is it DIP that really matters for GPP? Or rather is it the shallow/deep water mixing flux. DIP is very low in the surface ocean where it is limiting. But there can still be high GPP if the flux from below is high enough driving by extensive recycling. This raises an important point — you appear to be modeling marine primary productivity levels including both GPP and NPP. Please provide those values so we can see if they are reasonable. Also, what happens to deep-ocean O2 concentrations as you lower GPP and NPP at low DIP? Will we lose our anoxic zones? Or do you just have pO2 change to compensate — i.e., you are keeping dissolved ocean O2 the same.

Complex model —provide the estimated C/P ratios for the model run and compare to data? This would be quite helpful.

In the model we only calculate NPP. NPP is calculated using both DIP and ocean water mixing, which can be seen in equations C1 and C2 of Supplementary Table 9. We have also plotted NPP in Figure 5; as this figure shows, our NPP value is comparable to the modern value.

The DOA (degree of anoxia) in the deep ocean in the model is calculated assuming a water mixing term, a productivity term, an export term, and a surface oxygen concentration (from a pO_2 value and a temperature). If you lower NPP and thus organic matter burial, it will influence atmospheric pO_2 and the DOA.

We have now plotted C_{org}/P_{react} in Figures 2 and 3, Figure S4 and Figure S6. We didn't plot C_{org}/P_{org} as this term doesn't include CFA—the importance of which is the main topic of this study. But C_{org}/P_{react} and C_{org}/P_{org} values produced by our model are each comparable with empirical data from modern ocean sediments. For example, under anoxic conditions, C_{org}/P_{react} is approximately 180 (Fig. 2), which is within the range reported by Dale et al. (2016). Under oxic conditions, C_{org}/P_{react} is ~50, which is very similar to values measured at the FOAM site and is also comparable with those reported by Dale et al. (2016).

Figures

Figure 1: Do we have an external data on what a reasonable P regeneration percent should be? If we look at the modern calculated values, they are all predicted in the model to be about 50%. So I should expected, based on this, that typical C/P ratios of sediments (assuming they have chosen typical values) should be around 200-250. This seems correct to me. What about for the anoxic endmember? Is this consistent with data from those basins? If you look at C/P of 100 million year old sediments, are they different given you large predicted change in the regeneration rate?

We discuss the P regeneration percentage at greater length in the supplementary information. The P regeneration percent is ~60% to 75% in modern sites, meaning that the P burial efficiency is in the range of 25% to 40%. Our model results are comparable with that result.

As discussed above in response to the reviewer's previous point, in our 'standard' model run, under anoxic conditions, C_{org}/P_{react} is approximately 180 (Fig. 2), which is in the range compiled by Dale et al. (2016). Under oxic conditions with baseline conditions, C_{org}/P_{react} is ~50, which is the same value as what was been measured from the FOAM site and is also comparable with that reported by Dale et al. (2016). The C_{org}/P_{react} is inversely related to P burial efficiency. So, if the C_{org}/P_{react} value were different in ancient sediments, the regeneration rate would also be different.

Lastly, some anoxic sediment can have significant P enrichments (and, by inference, P accumulation rates; see Reinhard et al., Nature, 2018)—these are possible to achieve by varying TOC flux, sedimentation rate, $[Ca^{2+}]$, and other factors. In other words, although our baseline model runs give us expected P regeneration percentages, the model is flexible and is not parameterized to give *only* average values when there are strong shifts in depositional and other boundary conditions.

Figure 2: Is burial efficiency here just one minus the regeneration rate from figure 1? If so, perhaps use the same units. I would like to see the predicted C/P ratios for these curves as these can be more easily related to actual data to check whether what we are seeing is reasonable. Such a ratio is inherently calculated by the model.

The burial efficiency is indeed just one minus regeneration rate. We have, following the reviewer's suggestion, changed Figure 1 to use the same terms.

Thank you also for this useful second suggestion—we have now plotted $C_{\text{org}}/P_{\text{react}}$ values in the figure. These are indeed comparable to empirical data. For example, under anoxic conditions, $C_{\text{org}}/P_{\text{react}}$ is approximately 180 (Figure 2), which is within the range reported by Dale et al. (2016). Under oxic conditions, $C_{\text{org}}/P_{\text{react}}$ is ~50, which is the same value reported from the FOAM site and is also comparable with that reported by Dale et al. (2016).

Figure 3: Again I would provide all data in terms of a C/P ratio where possible. Additionally, something I find confusing is that total burial of P is a constant (it is just the weathering flux). So when the amount of CFA in sediments grow, what happens to the other sinks? I.e., where does the additional P come from or go to in the sediments as CFA concentrations change.

We have now plotted $C_{\text{org}}/P_{\text{react}}$ ratios in this figure. This figure is a sensitivity analysis for the diagenetic model rather than an output from the coupled model, so the other sinks, by definition, do not change as the organic flux is fixed. This will, however, change the P flux from the sediments to the water column, as is reflected by the changes in burial efficiency shown in Figure 2.

Figure 4: The model is built on the FOAM site, which is not a deep sea sediment. Is this an appropriate comparison? Why not use deep-sea sediments for the parameterization. Please break down the PCFA/Preactive based on the cores they came from and basins. It would be nice to make sure there is not an aliasing in the record. Finally, please show these a C/P ratios. I would like to make sure those change as would be predicted.

We actually gauged the utility of the model through comparison to both a shallow-water site (FOAM) and a deep-water site (ODP 1226) (and we have tried to make this more clear in the main text of the resubmitted version of the manuscript). We have parameterizations for both shallow-water and deep-sea sediments, as discussed in the Methods and Supplementary Information. The model results in Figure 4, for example, are from deep-sea sediments and demonstrate that the comparison is indeed appropriate.

We also now show $P_{\text{CFA}}/P_{\text{reactive}}$ for different ocean basins. There are, in total, 17 cores (shown in Table S7)—however, as we are sure the reviewer would agree, these are too much data to plot individually in a single figure.

Unfortunately, we have not been able to find organic carbon data for the same samples from which P speciation data have been reported. The organic carbon concentration was apparently not measured in tandem with P species by those who originally collected the core and published these results. This means that, the available data for C_{org} and P_{react} are mismatch in age. However, we are able to calculate the $C_{\text{org}}/P_{\text{react}}$ ratio using the 2 myr bin average value of C_{org} and P_{react} , although the existing data on C_{org} concentration could have a large error as the concentration of organic carbon is very low in deep sea sediments. We have included this figure as Fig. S6. The data has a large scatter (Fig. S6), but it is not at odds with our model results.

Figure 5 Is the pO2 actual percent (i.e. taking into account 0.8 bars of N2) or is it a pressure.

This is percent by volume—we have clarified this point in the figure caption.

Figure 6: Again, it would be useful to see a C/P ratio as this is something that is measurable and presumably predicted by the model given absolute corg burial fluxes are being used to determine pO₂.

We have removed this figure, following the suggestion of Reviewer #1.

Respectfully submitted (on behalf of all authors),

Noah Planavsky

Reviewers' comments:

Reviewer #1 (Remarks to the Author):

The authors have agreed with all of my comments and have made the changes or clarifications I have requested. So I am happy to recommend publication in this form, given that the editor remains interested.

I apologize if this review has been antagonistic and I do regret the phrase "they now agree that Ca concentration is not the most important control". What I meant to convey is that they now agree that they can't make such a sweeping conclusion. I agree that absolutism is not helpful, and I remind the authors that my reviewing for this paper has improved this aspect of the work. For example, the section that has only just been removed on overturning 'theoretical paradigms' or 'traditional frameworks' that may not actually exist in the first place.

Reviewer #2 (Remarks to the Author):

The authors present a new numerical model to demonstrate that the concentration of Ca in seawater has a strong effect on CFA formation in marine sediments. Since CFAs contribute significantly to P burial, high seawater Ca promotes P removal from the ocean and induces a decline in ocean productivity and potentially the O₂ content of the atmosphere. These are important results that justify the publication of the manuscript in Nature Communications. While I support the main conclusions of the manuscript, I am somewhat concerned about some aspects of the model:

1. The model predicts that an increase in the organic matter flux to the sediment surface (JOC) induces a decrease in CFA formation (Fig. 2c). This is a counterintuitive result because the concentrations of pore water phosphate and the saturation state with respect to CFA should increase when more organic matter is deposited at the seabed. Previous models confirm that organic matter deposition promotes CFA formation (Tsandev et al., 2012). The authors should explain the unexpected behavior of their model and check whether these model results are really valid and representative over a wider range of the parameter space.
2. The model predicts that an increase in bottom water oxygen induces an increase in CFA formation (Fig. 2a). This is again a counterintuitive result because elevated oxygen values should promote P fixation in iron oxides and should lower the pH of the pore water due to the release of metabolic CO₂ during aerobic respiration. A lower pH should induce a decline in CFA saturation because the elevated proton concentrations should shift the acid-base equilibria such that carbonate (CO₃²⁻) and phosphate (PO₄³⁻) concentrations decline. The authors should again explain the unexpected behavior of their model and check whether these model results are really valid. It remains unclear whether the unexpected model behavior shown in Fig. 2a and 2c is representative for a wide range of the parameter space. The authors should add a 3-D plot of their CFA model results to the supplementary information where the organic matter flux to the sediment surface and the oxygen concentration in bottom waters are varied simultaneously to explore the model behavior over a wider parameter space.
3. The legend to Fig. 5 is unclear. It indicates that the broken lines represent model results with variable Ca forcing while solid lines represent model outcomes with variable bioturbation. In contrast to the results explained in the text and shown in Fig. 3, the effects of Ca seem to be much smaller than those of bioturbation. Is this a printing error? Did you confuse Ca and bioturbation in the legend?
4. While I appreciate the attempt to simulate CFA solubility using a clearly defined ion activity product (IAP, Eq. 4), I do have a problem with the concept expressed in Eq. (5) where the solubility constant (K) is assumed to increase with the dissolved carbonate ion activity (aCO₃). This is a somewhat strange approach because the carbonate ion activity affects both IAP and K, i.e. an increase in aCO₃ leads to a rise in IAP and K and it remains unclear how these changes affect the saturation state. More importantly, the previously proposed aCO₃ dependency of the solubility constant relates to the carbonate content of the solid phase (CFA) rather than the aCO₃ of the solution (Jahnke, 1984) whereas the model presented in Eq. (5) seems to propose that the solubility increases with the aCO₃ of the solution. This is potentially a serious error. It may explain some of the unexpected behavior of

the model (Fig. 2). The authors should remove the a_{CO_3} dependency of the solubility constant (Eq. 5) and use a constant solubility for CFA that is representative for the CFA stoichiometry applied in Eq. (4). They should re-do the simulations to explore how this rather fundamental change affects the model results and/or present convincing arguments supporting the validity of their approach.

References

Jahnke, R. A., 1984, The synthesis and solubility of carbonate fluorapatite: *American Journal of Science*, v. 284, no. 1, p. 58-78.

Tsandev, I., Reed, D. C., and Slomp, C. P., 2012, Phosphorus diagenesis in deep-sea sediments: Sensitivity to water column conditions and global scale implications: *Chemical Geology*, v. 330, p. 127-139.

Reviewer #3 (Remarks to the Author):

This is my third time reviewing this paper (I am tracked as reviewer #3). The authors have worked hard to address my comments from both the first and second reviews. This paper will have impact in earth science by linking the calcium cycle to the phosphorus cycle (which in turn is key to global and net primary productivity, atmospheric O_2 , etc). This linkage is novel. I recommend publication.

Reviewers' comments:

Reviewer #1 (Remarks to the Author):

The authors have agreed with all of my comments and have made the changes or clarifications I have requested. So I am happy to recommend publication in this form, given that the editor remains interested.

Thank you for this positive review, and thoughtful previous reviews of this manuscript.

Reviewer #2 (Remarks to the Author):

The authors present a new numerical model to demonstrate that the concentration of Ca in seawater has a strong effect on CFA formation in marine sediments. Since CFAs contribute significantly to P burial, high seawater Ca promotes P removal from the ocean and induces a decline in ocean productivity and potentially the O₂ content of the atmosphere. These are important results that justify the publication of the manuscript in Nature Communications. While I support the main conclusions of the manuscript, I am somewhat concerned about some aspects of the model:

Thank you for this positive review, and the support of the publication of this manuscript. We have revised the manuscript following your suggestions, which are very helpful for the improvement of our manuscript.

1. The model predicts that an increase in the organic matter flux to the sediment surface (JOC) induces a decrease in CFA formation (Fig. 2c). This is a counterintuitive result because the concentrations of pore water phosphate and the saturation state with respect to CFA should increase when more organic matter is deposited at the seabed. Previous models confirm that organic matter deposition promotes CFA formation (Tsandev et al., 2012). The authors should explain the unexpected behavior of their model and check whether these model results are really valid and representative over a wider range of the parameter space.

This is a great question. We have added a new multiparameter plot to better show the influence of varying parameters on CFA burial (Fig. 2 and Fig. S4-S6). This plot does a better job of illustrating that CFA burial doesn't actually have a simple linear relationship with the organic matter flux (Fig. 2 and Figs. S4-S6). CFA burial can be positively correlated with organic flux when the organic flux is low. However, the relationship can reverse when there is a very high organic flux (Fig. 2 and Fig. S6). This makes sense given that there are multiple factors that control the saturation state of CFA that are affected by the extent of organic matter loading. Foremost, a sharp drop in porewater pH, which controls P speciation (the degree of phosphate protonation), near the sediment water interface can inhibit CFA formation and lead to a drop in the rate and extent of CFA formation. This same relationship was not observed in previous studies because the pH was not simulated (Tsandev et al., 2012). However, unlike other environmental forcings such as bioturbation, organic matter flux and bottom-water oxygen levels, seawater Ca concentrations always positively correlated with CFA burial (Fig. 2 and Figs. S4-S6), due to its direct effect on the saturation state of CFA. We have clarified these points in lines 88-93 and lines 101-104.

2. The model predicts that an increase in bottom water oxygen induces an increase in CFA formation (Fig. 2a). This is again a counterintuitive result because elevated oxygen values should promote P fixation in iron oxides and should lower the pH of the pore water due to the release of metabolic CO₂ during aerobic respiration. A lower pH should induce a decline in CFA saturation because the elevated

proton concentrations should shift the acid-base equilibria such that carbonate (CO_3^{2-}) and phosphate (PO_4^{3-}) concentrations decline. The authors should again explain the unexpected behavior of their model and check whether these model results are really valid. It remains unclear whether the unexpected model behavior shown in Fig. 2a and 2c is representative for a wide range of the parameter space. The authors should add a 3-D plot of their CFA model results to the supplementary information where the organic matter flux to the sediment surface and the oxygen concentration in bottom waters are varied simultaneously to explore the model behavior over a wider parameter space.

This is again a very great point. And thank you for the suggestion of 3-D plot, which we have added as Fig. S3. Similar to JOC, bottom water oxygen also does not have a simple relationship with CFA burial (Fig. S6). At low JOC ($0.5 \text{ mol m}^{-2} \text{ yr}^{-1}$), CFA burial increase with an increase in oxygen level when $[\text{O}_2]_{\text{BW}}$ is less than 20uM, but the relationship reverses when $[\text{O}_2]_{\text{BW}}$ is higher than 20uM (Fig. S6a). Bottom water oxygen also affects several factors that control the saturation state of CFA. As what you said, elevated oxygen level could have a negative effect on CFA burial due to its effect on pH. On the other hand, high oxygen level will promote the formation of iron oxides, which impedes the diffusion of dissolved inorganic phosphate (DIP) into seawater as it sequesters substantial phosphate (PO_4^{3-}) near the sediment-seawater interface. The reaction of iron oxides with H_2S at depth release the trapped phosphate (PO_4^{3-}), promoting CFA burial. We have modified lines 88-93 to try to make these points.

3. The legend to Fig. 5 is unclear. It indicates that the broken lines represent model results with variable Ca forcing while solid lines represent model outcomes with variable bioturbation. In contrast to the results explained in the text and shown in Fig. 3, the effects of Ca seem to be much smaller than those of bioturbation. Is this a printing error? Did you confuse Ca and bioturbation in the legend?

Sorry for the carelessness. Yes, we made a mistake on the legend. We have revised that.

4. While I appreciate the attempt to simulate CFA solubility using a clearly defined ion activity product (IAP, Eq. 4), I do have a problem with the concept expressed in Eq. (5) where the solubility constant (K) is assumed to increase with the dissolved carbonate ion activity ($a\text{CO}_3$). This is a somewhat strange approach because the carbonate ion activity affects both IAP and K, i.e. an increase in $a\text{CO}_3$ leads to a rise in IAP and K and it remains unclear how these changes affect the saturation state. More importantly, the previously proposed $a\text{CO}_3$ dependency of the solubility constant relates to the carbonate content of the solid phase (CFA) rather than the $a\text{CO}_3$ of the solution (Jahnke, 1984) whereas the model presented in Eq. (5) seems to propose that the solubility increases with the $a\text{CO}_3$ of the solution. This is potentially a serious error. It may explain some of the unexpected behavior of the model (Fig. 2). The authors should remove the $a\text{CO}_3$ dependency of the solubility constant (Eq. 5) and use a constant solubility for CFA that is representative for the CFA stoichiometry applied in Eq. (4). They should re-do the simulations to explore how this rather fundamental change affects the model results and/or present convincing arguments supporting the validity of their approach.

Thank you very much for this suggestion. Jahnke (1984) did suggest a relationship between solubility constant (K) with the dissolved carbonate ion activity ($a\text{CO}_3$). We followed that relationship in the previous runs. However, Jahnke (1984) also said that this relationship is just a “best guess” and speculative. Thus, we have also used the same stoichiometry of CFA and a fixed $K_{\text{spCFA}} (10^{-99.7})$ without a relationship with carbonate activity in a series of runs (Figs. S5, S6 and S10). We found that this change only has a subtle influence on the model results (Fig. 2, Fig. 5, Fig. S5 and Fig. S10), so although this aspect of CFA formation is not all that well constrained it does not influence on our conclusions. We have added relative statements making these points in lines 241-244.

References

Jahnke, R. A., 1984, The synthesis and solubility of carbonate fluorapatite: American Journal of Science, v. 284, no. 1, p. 58-78.

Tsandev, I., Reed, D. C., and Slomp, C. P., 2012, Phosphorus diagenesis in deep-sea sediments: Sensitivity to water column conditions and global scale implications: Chemical Geology, v. 330, p. 127-139.

Reviewer #3 (Remarks to the Author):

This is my third time reviewing this paper (I am tracked as reviewer #3). The authors have worked hard to address my comments from both the first and second reviews. This paper will have impact in earth science by linking the calcium cycle to the phosphorus cycle (which in turn is key to global and net primary productivity, atmospheric O₂, etc). This linkage is novel. I recommend publication.

Thank you for this positive review, and thoughtful and constructive previous reviews of this manuscript.

REVIEWERS' COMMENTS:

Reviewer #2 (Remarks to the Author):

The paper is now in very good shape. However, I am still not happy about the treatment of CFA solubility in the model. Equations 4 and 5 in the main text imply that an increase in the concentration of dissolved carbonate ions in porewater ($[\text{CO}_3^{2-}]$) causes a decrease in the rate of CFA formation. This counterintuitive behavior arises because the solubility of CFA (K_{spCFA}) rapidly increases with $[\text{CO}_3^{2-}]$ (Eq. 5) such that the saturation state (Ω_{CFA}) decreases with $[\text{CO}_3^{2-}]$ (Eq. 4).

I agree that the solubility of CFA increases with the carbonate ion content of CFA (SCO_3). It is also likely that SCO_3 somehow increases with $[\text{CO}_3^{2-}]$. However, the relationship between these variables is complicated and not fully understood.

The authors apply a constant CFA stoichiometry in Eq. 4. Thus, they ignore the increase in SCO_3 that may be induced by a rise in $[\text{CO}_3^{2-}]$ even though this change in CFA composition drives the increase in CFA solubility expressed via Eq. 5. This is clearly not a consistent approach.

Moreover, the assumption that CFA does not dissolve when $\Omega_{\text{CFA}} < 1$ (lines 243-244) is not valid from a thermodynamic perspective. It is in fact possible that CFA formed at depth dissolves when it is transported to the surface via e.g. bioturbation. The assumption may be valid if the kinetics of CFA dissolution is extremely slow but I am not aware of any data supporting this assumption.

The additional simulations with constant K_{spCFA} show that the $[\text{CO}_3^{2-}]$ -dependency assumed in Eq. (5) has no strong effect on the model results possibly because of the low carbonate content of CFA defined in Eq. 4. Nevertheless, I would like to suggest that the authors either change their concept or at least clearly highlight the consequences, uncertainties and inconsistencies of their approach.

Your manuscript entitled "The role of calcium in regulating marine phosphorus burial and atmospheric oxygenation" has now been seen again by a referee, whose comments appear below. In light of their advice I am delighted to say that we are happy, in principle, to publish a suitably revised version in Nature Communications under the open access CC BY license (Creative Commons Attribution v4.0 International License).

As you will see from the remaining reviewer's comments, there are some lingering concerns about CFA solubility in the model. Because I did not want to extend this process any longer, I reached out to the referee and asked for more clarification and a clear path forward regarding how to specifically address these final concerns. The reviewer replied that their concerns could be addressed by adding in a few caveats to the paper. In more detail, they replied (I quote):

Consequences: Equations 4 and 5 in the main text imply that an increase in the concentration of dissolved carbonate ions in porewater causes a decrease in the rate of CFA formation. This counterintuitive behavior arises because the solubility of CFA (K_{spCFA}) rapidly increases with $[CO_3^{2-}]$ (Eq. 5) such that the saturation state (Ω_{CFA}) decreases with $[CO_3^{2-}]$ (Eq. 4).

Uncertainties: The solubility of CFA increases with the carbonate ion content of CFA (SCO_3). It is also likely that SCO_3 somehow increases with $[CO_3^{2-}]$. However, the relationship between these variables is complicated and not fully understood.

Inconsistencies: In Eq. 4 we ignore any increase in SCO_3 (change in CFA stoichiometry) even though this change in CFA composition drives the increase in CFA solubility expressed via Eq. 5. (End quote)

We therefore invite you to revise your paper one last time to address these remaining concerns. At the same time we ask that you edit your manuscript to comply with our format requirements and to maximise the accessibility and therefore the impact of your work.

- We have added the caveats to lines 264-270 the main text.

In this parametrization, carbonate ion activity influences the saturation state of CFA through its effect on both IAP and K_{sp} . We have also used the same stoichiometry of CFA and a fixed K_{spCFA} ($10^{-99.7}$) without a relationship with carbonate activity in a series of runs (Supplementary Figures 5, 6 and 10) and found that this change only has a subtle influence on the model results (Figs. 2 and 5, and Supplementary Figures 5 and 10). It is also possible that stoichiometric abundance of CO_3^{2-} in CFA correlates with $[CO_3^{2-}]$ of porewater¹¹. However, the relationship between the solubility of CFA, the stoichiometric abundance of CO_3^{2-} in CFA and $[CO_3^{2-}]$ of porewater are not fully understood¹¹. These uncertainties may influence the relationship between $[CO_3^{2-}]$ of porewater and CFA formation, which is currently a negative correlation (Fig. 3). Further, these will not influence our main conclusions as they do not significantly influence the relationship between $[Ca^{2+}]$ and the saturation of CFA (Ω_{CFA}).

- We have also modified the formats of the manuscript following the checklists.